# Foreshocks and Short-Term Hazard Assessment to Large Earthquakes using Complex Networks: the Case of the 2009 L'Aquila Earthquake

E. Daskalaki[1,2], K. Spiliotis[2], C. Siettos[2], G. Minadakis[1], G. A. Papadopoulos[1]

[1]Institute of Geodynamics, National Observatory of Athens, Athens, GR-11810, Greece
[2]School of Applied Mathematics and Physical Sciences, National Technical University of Athens, GR-15780, Politechnioupoli, Zografos, Athens

*Correspondence to:* C. Siettos (ksiet@mail.ntua.gr)

**Abstract.** The monitoring of statistical network properties could be useful for the short-term hazard assessment of the occurrence of mainshocks in the presence of foreshocks. Using successive connections between events acquired from the earthquake catalogue of INGV for the case of the L'Aquila (Italy) mainshock ($M_w = 6.3$) of 6th April 2009, we provide evidence that network measures, both global (average clustering coefficient, small-world index) and local (betweenness centrality) ones, could potentially be exploited for forecasting purposes both in time and space. Our results reveal statistically significant increases of the topological measures and a nucleation of the betweenness centrality around the location of the epicenter about two months before the mainshock. The results of the analysis are robust even when considering either large or off-centered the main event space-windows.

## 1 Introduction

Seismicity is a 3-D complex process evolving in the heterogeneous space, time and size domains. Since the birth of the science of seismology about 130 years ago, the underlying statistical properties of seismicity have attracted increasingly great interest (see e.g. a review in Utsu, 2002), enhancing our understanding of the complex physical mechanisms that cause earthquakes. Over the years, several models have been proposed for the description of seismicity. For example, the random walk model (Lomnitz, 1974) was introduced to describe on-clustered background seismicity. However, space-time earthquake clusters deviate significantly from randomness. In fact, the pioneering work of Omori (1894), extended later by others (e.g. Utsu, 1962a, b, Utsu et al., 1995) revealed the strong clustered nature of aftershock sequences following large mainshocks. Aftershocks decay with time in a power-law mode, the so-called Omori law. However, as the decay of aftershocks very often deviates from the simple power-law, the statistical model of Epidemic Type Aftershock Sequences (ETAS) was introduced (Ogata, 1998) to describe the complex pattern of aftershocks time distribution (Zhuang, 2012). On the other hand, a physical approach based on the rate- and state-model of fault friction was also introduced (Dieterich, 1994).

In some occasions, short-term foreshocks preceding mainshocks by hours, days or up to a few months were reported. It was found that the number of foreshocks generally increases with the inverse of time (Mogi, 1962, 1963a, b; Papazachos, 1975; Kagan and Knopoff, 1978; Jones and Molnar, 1979; Hainzl et al., 1999; Main, 2000; Papadopoulos et al., 2010). Therefore, foreshocks may provide time-dependent information which may lead to a more robust estimation of the probability for the

occurrence of future strong mainshocks (e.g. Agnew and Jones, 1991). However, some mainshocks are preceded by foreshocks while others do not. Long-term accelerating foreshock activity has been also described (for a thorough review see in Mignan, 2011). A very early case in the Hellenic Arc was studied by Papadopoulos (1988), while models for the long-term accelerating seismicity were further developed by others (e.g. Bufe and Varnes, 1993).A recent revision of these models has been proposed in order to cope with some previous limitations (De Santis et al., 2015). Another major type of

space-time seismicity clusters, termed earthquake swarm, is characterized by a gradual rise and fall in seismic moment release but it is lacking a foreshocks-mainshock-aftershocks pattern (e.g. Yamashita, 1998; Hainzl, 2004; Hauksson et al., 2013). Although seismic swarms are abundant in volcanic and geothermal fields as well as in areas of induced seismicity, caused by fluid-injection, mining or gas recovery, they are not unusual in purely tectonic settings.

Over the last years, the complex network theory has provided a new insight and perspective in analyzing seismicity patterns

(Abe and Suzuki, 2004, 2007; Baiesi and Paczuski, 2004, 2005; Barrat et al., 2008; Daskalaki et al., 2014). This line of research is motivated by the concept of self-organized criticality (Bak, 1996) which models structural phase transitions from random to scale-free spatial correlations between seismic events (e.g. Sammis and Sornette, 2002). Yet, the generalization and reliability of the outcomes of this relatively new approach remains an open question. Within this framework the so-called Visibility Graph (VG) method maps time series into networks or graphs, which is converting dynamical properties of

time series in topological properties of networks, can to identify possible precursory signatures (Telesca et al., 2012; 2015)

In this paper, we exploited the tools of complex network theory to identify potential spatio-temporal foreshock patterns that could add value to the short-term earthquake forecasting or hazard assessment. We focused on the case of the shallow, strong ($M_w = 6.3$), lethal mainshock which ruptured the Abruzzo area, central Italy, on April 6th, 2009, at UTC 01:32:39, with an

epicenter situated at 42.42°N, 13.39°E (Fig.1), near the city of L'Aquila (http://terremoti.ingv.it/it/). Fault-plane solutions for the mainshock and aftershocks are consistent with predominantly normal faulting striking NW-SE and dipping to SW, with a minor right-lateral component (e.g. Walters et al., 2009).The mainshock was followed by abundant aftershocks with the major events occurring on the 7th of April ($M_w = 5.6$) and on the 9th of April ($M_w = 5.4$).Immediately following the mainshock, Marzocchi and Lombardi (2009) began producing daily aftershock forecasts based on a stochastic model that

combines the G-R (Gutenberg and Richter, 1944) distribution and space-time power-law decay of triggered shocks.

We selected to study the L' Aquila seismic sequence since the mainshock was preceded by abundant foreshocks (Papadopoulos et al., 2010; De Santis et al., 2011), thus allowing to test topological metrics, which are in use in the complex network theory. Such metrics are potential tools for the investigation of short-term precursory seismicity patterns. The aim is to show if and how the exploitation of network theory metrics can independently add value to short-term seismic hazard

assessment in the presence of foreshocks Therefore, before the implementation of selected topological metrics, in Section 2 we examined further the seismicity patterns that preceded the L' Aquila mainshock by mainly focusing in statistics of the foreshock sequence in space, time and size (magnitude) domains.

Here, we adopted the term seismic hazard assessment referring to "forecasting"/ "hindcasting" (Evison, 1999; Bormann, 2011), i.e. a warning that a mainshock would probably happen within a specific region in short-term, instead of the term "prediction" which contains a much stronger statement that a mainshock will deterministically happen. In this sense, earthquake forecasts are prospective probabilistic statements specifying the likelihood that target events will occur in space-time subdomains. In a time-dependent forecast, the probabilities $P(t)$ depend on the information $I(t)$ available at time $t$ when the forecast is made (Jordan et al., 2011). A thorough global review has shown that strategies of time-dependent hazard assessments for the earthquake forecasting could be established in a real-time framework (Jordan et al., 2011).

## 2 Seismicity patterns preceding the L'Aquila mainshock

Long-term seismicity analysis showed that the L'Aquila mainshock was preceded by seismic quiescence prevailing for about 40-50 years, thus very likely filled in a seismic gap (Barani and Eva, 2011). It was also suggested that the earthquake was hindcasted from a fault-based earthquake rupture model (Peruzza et al., 2011). During the two years before the event, no anomalous strainmeter signal larger than a few tens of nanostrains was visible but during the last few days, there was evidence of dilatancy of saturated rock over the earthquake causative fault, perhaps related to the presence of foreshocks (Amoruso and Crescentini, 2010). However, around a year before the mainshock possible effects due to fluid migration was found from magnetic data analyses (Cianchini et al., 2012). The non-extensivity parameter $q$ increases in the seismic interval from 30 March 2009 to 6 April 2009 before the occurrence of the L'Aquila event, indicating an increase in the degree of out-of-equilibrium state before the occurrence of this strongest event (Telesca, 2010).

In the short-term, the mainshock was preceded by a foreshock sequence which developed in two main stages (Papadopoulos et al., 2010). Namely, a posteriori analysis of the INGV catalogue data (http://bollettinosismico.rm.ingv.it) showed that from the beginning of 2006 up to the end of October 2008 the activity was relatively stable and remained in the state of background seismicity (seismicity rate $r$=1.14, $b$=1.09; where $b$ is the slope of the magnitude-log frequency or G-R relation). In the earthquake magnitude domain, the magnitude-frequency (or G-R) relation (Gutenberg and Richter 1944) reads as $logN = a - bM$. This relation, which describes the power-law decrease of the number of events with the increase of magnitude, has been found to apply in both clustered and non-clustered types of seismicity (see review in Utsu 2002); $N$ is the cumulative number of events of magnitude equal to or larger than $M$ and $a$, $b$ are parameters determined by the data. By the end of October 2008 up to 26 March 2009, $r$ increased significantly to 2.52 indicating weak foreshock sequence; the $b$-value did not changed significantly. The weak foreshock sequence was spatially distributed within the entire rupture area determined by the aftershock spatial distribution (Papadopoulos et al. 2010, and their Fig. 1c). In the last 10 days before the mainshock, strong foreshock signal became evident in space with dense epicenter concentration in the hanging-wall of the

Paganica fault (Fig. 1), in time ($r$=21.70 events/day) and in size ($b$=0.68). It has been also suggested by De Santis et.al., 2010, that the foreshock sequence was produced by a physical process dominated by a strong chaotic component as recognized by the accelerated strain release.

We examined further the seismicity evolution in the time-space-size domains before the L' Aquila mainshock as illustrated in Fig. 2. The earthquake catalogue of INGV (http://legacy.ingv.it/roma/reti/rms/bollettino/index.php?lang=en) was implemented in our seismicity analysis with data covering the time period from 1[st] January 2008 to 30[th] June 2009.The earthquake catalogue was tested for data completeness on the basis of the G-R diagram and the magnitude cut-off $M_c = 1.3$ was selected, which is consistent with previous findings (e.g. Papadopoulos et al., 2010). We have considered the earthquake catalogue of the area without threshold in the focal depth. However, the majority (nearly 97%) of the events have focal depth less than 30 km. The remaining may have depth up to about 45 km but we did not remove them from the catalogue allowing for some error to be involved in the depth determination. As a consequence, the data set that we used practically represents the shallow seismogenic layer.

Figure 2a shows the cumulative number of earthquake events within a circle of radius of 30 km from the L' Aquila mainshock epicenter. The dramatic increase of the seismicity rate in about the last 3 months before the mainshock of 6 April 2009 is evident. Particularly, in the last 10 days the seismicity rate increased at significance level 99% according to the z-test, as it was already shown in a previous paper (Papadopoulos et al., 2010). We tested this pattern for several radii gradually decreased up to 5 km or increased up to 100 km from the mainshock epicenter. It was found that the pattern was still significant at level 95%. However, for radii of less than 5 km or more than 100 km the pattern gradually loosed significance due either to the decreasing number of events involved or to the inclusion of increasing number of events associated with other seismogenic sources.

With the increase of the seismicity rate in the last 10 days, that is during the strong foreshock stage, the mean distance of foreshock epicenters from the mainshock epicenter decreased, being about 7 km (Fig. 2b). The time evolution of the parameter $b$ before (red) and after (blue) the L' Aquila mainshock showed also very distinct patterns (Fig. 2c). Before the foreshock sequence it varied from 0.9 to 1.2, that is it was close to the theoretical value of 1. During the weak foreshock activity $b$ remained also close to 1. About two months before the mainshock it dropped gradually reaching values of less than 0.7 in the last 10 days. We applied the Utsu (1992) test for testing the significance level, $p$, of the $b$-value variation. In the last 10 days it was found $p$=0.002, which means significance level of 0.998 in the $b$-value drop.

## 3 Topological metrics

Building up on previous efforts (Abe and Suzuki, 2004, 2007; Baiesi and Paczuski, 2004, 2005; Barrat et al., 2008; Daskalaki et al., 2014), exploiting the arsenal of complex networks, we were able to independently investigate statistically significant changes in the underlying seismic network topology marked about two months before the mainshock. Our analysis was based on the earthquake catalogue of INGV.

We discretized the area under study (Fig. 1) into square cells with a side of 0.1°. Then, we tessellated the catalogue into successive overlapping sliding windows. For each sliding window, a network was constructed by linking cells successively in time, when seismic activity was observed within cells. Denoting by $t_i, t_{i+1}$ the time instances when two successive seismic events occur within $i$ and $j$ cells, respectively, then we link $i$ and $j$ cells($i \rightarrow j$), which represent nodes of the underlying network. This approach is based on the Abe and Suzuki (2004, 2007) approach. The use of the sliding window allows the tracing of the temporal changes in the network topology (Daskalaki et al., 2014). Here, we propagated the sliding window using two alternatives: either by a constant-time period (here, one day) or by constant number of seismic events (here set to 10 events). Within each sliding window, we measure the following properties of the underlying directed network: the small-world index ($SW$), the average clustering coefficient ($ACC$), the betweenness centrality ($BC$) and the mean degree of the underlying degree distribution (Newman, 2003; Albert and Barabasi, 2002; Costa et al., 2007; Fagiolo, 2007; see in Appendix for mathematical definitions). The clustering coefficient is the ratio between all directed triangles actually formed by node $i$ and the number off all possible triangles that node $i$ could form (Fagiolo, 2007) and consequently by averaging the $ACC$ measures the cliquishness (structure) of the network (Watts and Strogatz, 1998). Note that the $BC$ metric is a property that refers to a certain node in the network, while the $ACC$ and the mean degree are global network properties in the sense that they are obtained by averaging the corresponding properties over all the nodes of the network. It is known that nodes exhibiting high values of $BC$ are highly participating to the flow of information, including flow of energy, in the network. The small-world index is an important statistical measure that reveals how a network structure deviates from random structures that account for regular seismicity. Thus, the $SW$ index can be used to characterize phase transitions that mark the onset of relatively big changes in the underlying topology. The original definition comes from Humphries et al. (2006) and gives an efficient way of characterizing all together the structure of the network with respect to the relation between clustering coefficient and path length.

In order to detect statistically significant changes between the measures obtained from the emergent seismic networks and the ones resulting from consistent random network realizations, the following procedure was applied. Within each sliding window, we constructed an ensemble of 500 realizations of consistent random networks, i.e. random networks with the same number of nodes and with connectivity probability equal to the average degree (see appendix for definition) of the emerged seismic network divided by the number of nodes (Newman, 2003; Albert and Barabasi, 2002). For each of the 500 random network realizations, we computed the statistical measures mentioned above. We adopted as statistically significant the values that were above the 95% or below the 5% of the distributions calculated from the random networks.

In order to test the robustness of the outcomes of the analysis, we constructed networks using different values of sliding window lengths and shift steps as well as different sizes of centered or off-centered, with respect to the mainshock epicenter (42.42°N, 13.39°E), space-windows. Within a wide range of these values, the outcomes of the analysis were equivalent. As during the two years period before the mainshock, no anomalous strainmeter signal larger than a few tens of nanostrains was visible, it was suggested that the volume of the possible earthquake preparation zone was limited to less than 100 km$^3$ (Amoruso and Crescentini, 2010).This is also valid for the seismogenic zone as determined by the area covered by the cloud

of foreshocks and aftershocks (Fig. 1). Therefore, the space-windows were selected larger than the seismogenic area of the earthquake sequence, so that we would not miss any critical seismicity information.

## 4 Results

For our illustrations, we show the results obtained using a sliding window with a shift step of either constant-time of one day or of constant-number-of-events $N_s = 10$ and a space- window centered at the mainshock epicenter with two different radius, $R_1 = 1^o$ and $R_2 = 3^o$. The initial size of the sliding window was set to $N_0 = 100$. The total number of cells were 400 and 3420 for $R_1$ and $R_2$ radius, respectively. The resulting time-series of the statistical measures $ACC$ and $SW$ index in the period from 1 January 2006 to 30 June 2009 using the centered at the mainshock epicenter space-window with radius $R_1 = 1^o$ are shown in Fig.3. The time evolution of the network measures indicates the existence of two distinct phases/structures with an apparent in-between phase-transition initiating around two months before the mainshock of 6[th] April 2009 (Fig. 3). The phase-transition corresponds to the period where the weak stage of the foreshock sequence was developed. The initial phase pertained to the period from the beginning of the catalogue segment examined until about two months before the mainshock, while the second phase is associated with the aftershock sequence. In the initial phase, the network resembles a random graph whose topology is characterized by the $ACC$ and $SW$ indices within the statistically significant thresholds set by the random network realizations. In the aftershock period, our analysis revealed a statistically significant change in the emerged network structure. In particular, the average $ACC$ jumps up to higher values around 0.5 (and remains high) and the $SW$ index exceeds 10 indicating the strong small-world character of the underlying topology.

For larger space-windows of $R_2 = 3^o$ (Fig. 4), the results are, for any practical means, equivalent to the case of $R_1 = 1^o$. Of particular interest are the intermediate intervals between the above two distinct phases of network structures (around two months, and especially 10 days, before the mainshock). In terms of network topology interpretation, higher values of $ACC$ implie more clustered and organized seismicity pattern. Furthermore, within the period of two months, the $BC$ reveals a transition between the two phases, since the spatial distribution of $BC$ is localized to few nodes, which act as hubs. This means that the seismicity revolves around this particular point (Fig. 5 and 6). These hubs are the stopover between any two nodes and consequently, the distance between any two nodes is reduced. The seismicity transition which is captured in the network topology transition is also confirmed by the evolution of the $SW$ index: it remains within the expected range of values set by the random configuration until two months before the mainshock; then it passes the threshold and remains above it for the whole period afterwards until the mainshock, thus marking the second foreshock phase. In this phase, the emergent topology is characterized by statistically significant more clustered networks, with profound small-world characteristics. It is worthy to mention that the $ACC$ measure depends on the mean degree. Fig. 7 shows the evolution of the mean degree for the case of sliding windows with a constant time shift of 1 day. Clearly, the mean degree does not exceed the statistically significant bounds of 5% and 95% in the underlying seismic network topology.

Yet, an important question that naturally emerges is the following: is it possible to "forecast" the spatial location of a probable large earthquake from the identification of phase changes that have arisen on the topology of the underlying emerged networks? To respond to such a crucial question we computed the $BC$ for each node, trying to identify hubs that could serve as potential epicenter locators. Figures 5 and 6 show snapshots of the $BC$ map for the space-window of $R_1$ and

$R_2$ radii, respectively, around the mainshock epicenter, before (Fig. 5a-h and 6a-h) and after (Fig. 5i and 6i) the mainshock. Interestingly, it is shown that during the period until the $20^{th}$ of January 2009, the $BC$ values appear randomly dispersed in space. However, about two months before the mainshock and in particular from the 30th of January 2009 a different pattern emerged. Specifically, at the cell of the mainshock epicenter, there was a persistent appearance of large values of the $BC$ throughout the entire period from the end of January 2009 until the occurrence of the mainshock (Fig. 5a-h). This is clearly

illustrated in Fig.5j-k depicting that the cumulative $BC$, which is computed at the node of the epicenter (i.e. the mainshock cell, denoted by $CBC^*$), increases sharply and monotonically, approximately two months before the date of the mainshock, without any intermediate plateaus. Thus, there was a centralization of the $BC$ distribution at the cell of the epicenter. This behavior is unique and characteristic just for the node related to the mainshock epicenter. This observation along with the drop of the $b$-value (e.g. Papadopoulos et al., 2010) indicates a stress increase close to the mainshock epicenter, thus

underlying the physical link between the centralization of the $B$C distribution and the mainshock nucleation process. The $BC$ of all other nodes do not change their values significantly as presented here. The $BCs$ of all other nodes of the network reach plateaus, i.e. their (cumulative) changes during the two-month period before the main event is negligible compared to the $CBC^*$. The corresponding $CBC^*$ continues to increase after the main event until late June, a behavior which is related to the aftershock activity. The above pattern characterizes both areas determined by radii $R_1$ and $R_2$.

In order to test if these results were sensitive to the selection of the center of the space-window used for the construction of the network, we repeated the analysis using off-the epicenter-centered space-windows. Off-the-epicenter analysis was also employed resulting to equivalent outcomes. In Fig.8, we depict the $BC$ map for the space-window with radius $R_1$ centered at the point 42.42°N, 13.39°E, which is about 80 km south from the mainshock epicenter. As it is shown, the results are qualitatively equivalent with those of Fig.5 and 6, i.e. with the results obtained when using a space-window centered at the

epicenter of the mainshock. Finally, in Fig.9, we illustrate snapshots of seismic networks overlaid on the $BC$ contours for the periods: (a) 26 September to 10 October 2008, (b) 14 to 27 January 2009, (c) 20 March to 4 April 2009, and, (d) 1 to 16 April 2009.

## 5 Discussion

The drastic increase of the seismicity rate is a common feature in foreshocks, swarms and aftershocks. Therefore, such

seismicity clusters are traditionally considered as retrospective designations: they can only be identified as such after an earthquake sequence has been completed (Jordan et al., 2011) given that certain criteria for the discrimination of foreshocks from other types of space-time seismicity clusters are needed (Ogata, 1998). Although this is in general true, the

retrospective analysis of the 2009 L' Aquila foreshock sequence showed that in a scheme of regular, daily statistical seismicity evaluation, the ongoing state of weak foreshock activity would be detectable in about one or two months before the mainshock (Papadopoulos et al., 2010). Then, the strong foreshock signal, being evident in the space, time and size domains could be detectable a few days before the mainshock. The presence of foreshocks, as states of elevated seismicity

with respect to background seismicity level, could be also suggested by independent approaches, such as Poisson Hidden Markov Models (Orfanogiannaki et al., 2014). The spatial organizations of foreshocks as a tool to forecast mainshocks has been positively examined (Papadopoulos et al., 2010; Lippiello et al., 2012; see also results in section 2). In the size domain, the drop of the $b$-value in foreshock sequences (Papadopoulos et al., 2010; see also results in the present paper) is of crucial importance for the real-time recognition of foreshock activities based on seismicity analysis. However, whatever is the

method to detect an ongoing foreshock activity, the magnitude of the forthcoming strong earthquake, $M_o$, would remain uncertain although preliminary results from a set of few but well-determined foreshock sequences around the globe tentatively indicated that $M_o$ is a function of the area covered by the foreshock activity (Papadopoulos et al., 2015). Sugan et al. (2014) analysed continuous waveforms from 10 broadband seismic stations in a 60 km radius from the epicenter and for approximately 3 months before the mainshock. They found that the relocated foreshocks mostly activated the deepest

northern portion of the L'Aquila main fault plane in the 3 months period preceding the $M_w = 6.3$ event. The $M_w = 6.3$, 6 April 2009 L'Aquila earthquake was preceded by a long suite of foreshocks, the largest one of magnitude $M_L = 4.1$ (30[th] of March at 13:38 (UTC)) marked the beginning of an abrupt temporal change in different seismic parameters, such as the b value (Papadopoulos et al., 2010; Sugan et al., 2014), the spatiotemporal distribution of the events (Telesca, 2010) and the P-to-S wave velocity ratio (Di Luccio et al., 2010; Lucente et al., 2010). Calderoni et al. (2014) state that before that event,

seismicity was concentrated to the north of the volume where the main shock nucleated. After the $M_L = 4.1$ event, rate and magnitude of foreshocks increased and seismicity migrated toward the main shock nucleation zone. Gulia et al. (2016) studied a circular area of 20 km radius, centered on the mainshock epicenter. They observed a foreshock sequence started three months before the mainshock, activating a region of about 10 km length.

Compared to the above studies our analysis provides an alternative way to describe the spatiotemporal precursory seismicity

changes. Thus, it is worth to mention that our method succeeded to determine the mark of onset of significant changes in seismicity when also considering an off-epicenter analysis (Fig.8). Based on the $BC$ measure, the identification of the spatial location of the epicenter two months before the main event was also possible.

One of the advantages of the complex network theory is that networks may be used to identify efficiently, within the nonlinear dynamics theory framework, phase-transitions that mark the onset of big changes in the underlying seismicity. The

wealth of statistical measures of the reconstructed network activity (such as the small-worldness, path-length, local and global clustering coefficient, betweenness centrality) offers many different views and tools for characterizing the underlying varying topology. We showed that key topological measures of the emerged seismic network constitute an independent tool for hazard assessment of the occurrence of the mainshock in the short-run. In this sense, the proposed approach looks promising as it could identify (retrospectively as all other methods until now) quite efficiently, about two months before the

mainshock, the location of the mainshock epicenter. Interestingly, in the a posteriori analysis of the 2009 L' Aquila seismic sequence, the betweeness centrality and its cumulative expression started to pinpoint the nucleation area of the forthcoming strong earthquake two months before its occurrence (look also De Santis et al. 2015).

Nevertheless, the detection of a seismicity anomaly in space and time by topological measures does not provide evidence on the seismicity style beforehand: it is designated only retrospectively. In fact, the foreshock style of seismicity becomes obvious only with the a posteriori knowledge that the anomaly concluded with a strong mainshock. Such knowledge, however, could be obtained from classic statistics beforehand on the basis of the 3-D space-time-size seismicity analysis. Furthermore, the role of the parameter $b$ variations is critical. Strong variations of $b$ across different stress regimes imply that this parameter acts as a stress meter that depends inversely on differential stress (e.g. Schorlemmer et al., 2005; Narteau et al., 2009 and references therein). Observations on seismic sequences have shown that $b$ usually drops and becomes significantly lower in foreshocks than in aftershocks or in background seismicity (Papazachos,1975; Jones and Molnar, 1979; Main et al., 1989; Molchan et al., 1999; Enescu et al,. 2001; Nanjo et al., 2012). This was also the case of L'Aquila (Papadopoulos et al., 2010; De Santis et al., 2011). Recently, Olami-Feder-Christensen (OFC) spring-block models and simulation experiments were utilized to bridge the macroscopic $b$-value to source mechanics, e.g. to elastic properties, as well as to stochastic structural heterogeneities in the source, thus modeling significant changes of $b$, e.g. during foreshocks, with a process of material softening (Avlonitis and Papadopoulos, 2014).

Hence, in view of the statistical and geophysical significance of $b$, it becomes interesting to investigate for alternative metrics, such as the topological ones that the network theory provides, in terms of spatio-temporal forecasting.

**6 Conclusions**

Utilizing complex network theory, we showed that key topological measures, such as the average clustering coefficient ($ACC$), small world index ($SW$) and the betweenness centrality ($BC$), could serve as potential indices for the short-term seismic hazard assessment. Of particular interest is the detection of forthcoming mainshocks in the presence of foreshocks.

In the case of foreshocks that preceded the L'Aquila (Italy) mainshock ($M_w = 6.3$) of 6[th] April 2009, statistically significant changes of the network topology as reflected by certain global measures, such as the $SW$ index and $ACC$, emanated simultaneously about two months before the mainshock. In this sense, the topological measures perform equivalently to classic statistics. However, a clear centralization of the $BC$ around the location of the mainshock epicenter appeared again about two months before the mainshock, persisting up to the mainshockoccurrenceThe recognition of the seismicity anomaly by topological measures, however, does not discriminate the seismicity style, i.e. foreshocks, swarms or aftershocks. The foreshock nature of the anomaly was detected only by the a posteriori knowledge that the earthquake sequence concluded with a mainshock. In seismology several branching type models have been used to identify space-time clusters, e.g. the epidemic-type aftershock sequence (ETAS) model (OGATA, 1998). Although it was tested for analyzing clustering features of foreshocks (e.g. ZHUANG and OGATA, 2006) no standard method has been introduced so far for the foreshock

recognition beforehand. This is also valid for the several techniques introduced for declustering earthquake catalogues (e.g. Jimenez et al., EPJST, 2009). However, the classic seismicity statistics yields possibilities for such discrimination beforehand thanks to the discriminatory power of the $b$ parameter which drops significantly during foreshocks. The drop of $b$ is not only of statistical but also of geophysical value. Investigating for an equivalent discriminatory topological measure is a

challenge.

The proposed approach holds promising regarding the identification of spatio-temporal patterns related to the underlying seismicity and thus could be potentially serve as alternative and/or complementary to well-established traditional statistical methods for short-term, time-dependent hazard assessment of earthquakes.

**Appendix**

Within each sliding window, we computed the following statistical properties of the emerged networks (Watts and Strogatz, 1998; Newman 2003; Albert and Barabasi 2002; Costa et al. 2007, Fagiolo, 2007):

a)      The mean degree. Generally the degree of a node $i$ is the number of edges connected to a node (Newman, 2003). In directed networks a node has both an in-degree and an out-degree, which are the numbers of in-coming and out-going edges respectively (Newman, 2003).Averaging over all the nodes of the network we get the mean degree.

b)      The average clustering coefficient ($ACC$), which reflects the average, the prevalence of clustered connectivity around individual nodes. The $ACC$ is defined as the mean value of all clustering coefficients $c_i$:

$$c_i = \frac{(A+A^T)^3_{ii}}{2[k_{tot}(k_{tot}-1)-2(A^2)_{ii}]}.$$

$A$ is the adjacency matrix of the network, $k_{tot}$ is the summation of inward and outward degrees, i.e. $k_{tot} = k_{in} + k_{out}$ and the parenthesis $()_{ii}$ indicates the main diagonal.The$c_i$is the ratio between all directed triangles actually formed by node $i$ and

the number off all possible triangles that node $i$ could form (Fagiolo, 2007). It is a natural extension of the Watts and Strogatz definition, since if it restricted to the undirected networks gives the same results.

The $ACC$ ranges from zero to one. Small values of $ACC$ of order $ACC = \frac{\bar{k}}{N}$ correspond to random network structures (encounter in normal seismicity periods).

c)      The small-world index($SW$)defined as:

$$SW = \frac{ACC_{rel}}{APL_{rel}}.$$

$ACC_{rel} = \frac{ACC}{ACC_{rand}}$ and $ACC_{rand}$ is the average $ACC$ computed from an ensemble of 1000 random network realizations (with the same number of nodes N and same mean $k_{tot}(i)$). $APL_{rel}$ stands for the relative average path length defined as:$APL_{rel} = \frac{APL}{APL_{rand}}$, where$APL$ is the average path length of the underlying seismic network defined by: $APL = \frac{\sum d_{i \to j}}{N(N-1)}.$

$d_{i \to j}$ is the shortest path between nodes $i$ and $j$and $APL_{rand}$ is the average path length computed from the ensemble of 1000

equivalent random networks. Small-world networks are formally defined as networks that are significantly more clustered

than random networks, yet have approximately the same characteristic path length as random networks. In the small-world topology $ACC \gg ACC_{rand}$ and $APL_{rand} \approx APL \rightarrow SW \gg 1$, meaning that high (abrupt) increment of $SW$ indicates the transition from normal to abnormal seismicity as it is reflected to network topology.

d)       The betweenness centrality ($BC$) defined as the fraction of all shortest paths in the network that pass through a given node. Bridging nodes that connect disparate parts of the network often have a high betweenness centrality. The betweenness centrality of a node $i$ is defined as:

$$BC(i) = \sum_{j \neq i \neq k} \frac{g_{jk}(i)}{g_{jk}}.$$

$g_{jk}(i)$ is the number of shortest path passing from node $i$, and $g_{jk}$ is the number of shortest paths between nodes $i$ and $j$. Higher values of $BC(i)$ indicates that the $i$ node acts as a central node influencing most of the other nodes in network. This means that this node (patch of space acts as a hub.

*Acknowledgements.*This is a contribution of the research project EARTHWARN of the Institute of Geodynamics,National Observatory of Athens

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

**Figures**

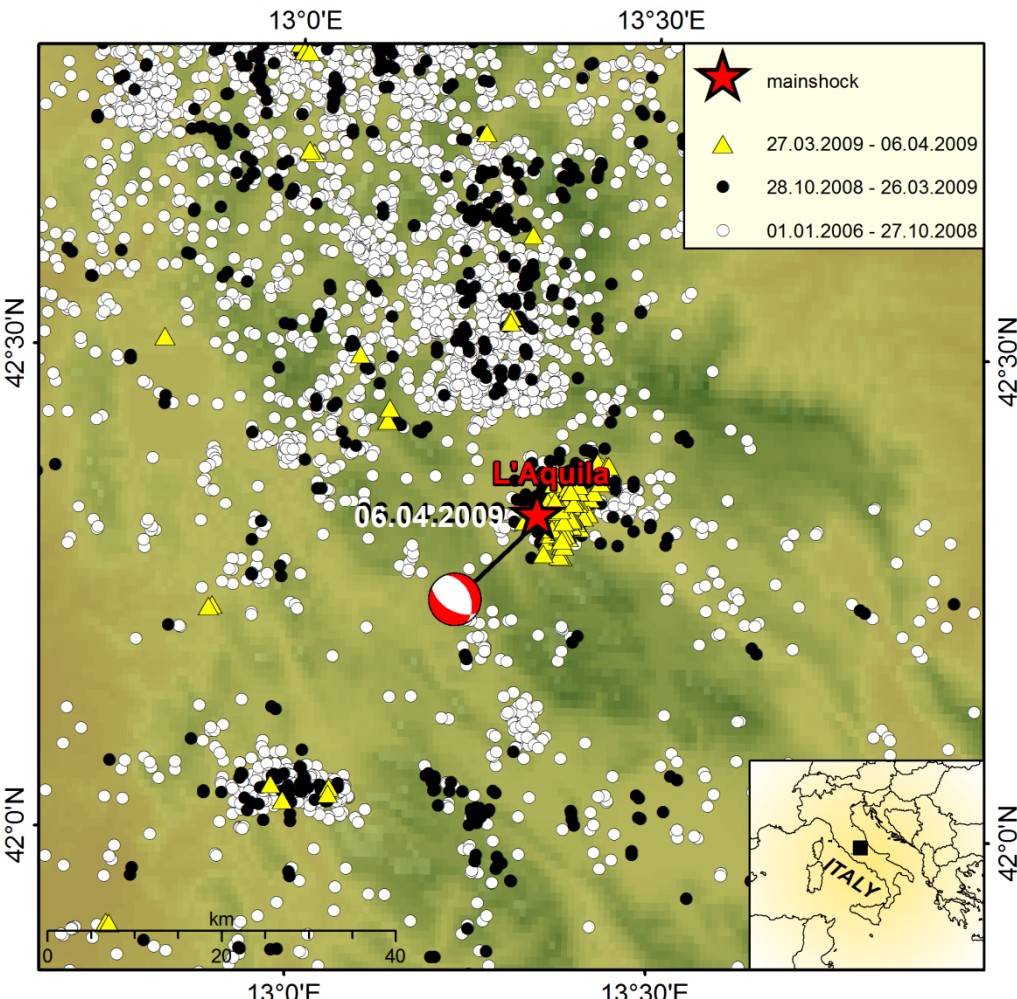

**Figure 1**. Epicentral distribution of earthquakes in the area of L'Aquila for the time interval extending from 01.01.2006 to 30.06.2009.The focal mechanism of the L' Aquila mainshock (star) was calculated by INGV. Note the dense concentration of foreshock epicenters close to the mainshock epicenter in the last 10 days preceding the mainshock occurrence, which indicates that foreshocks moved towards the mainshock nucleation area.

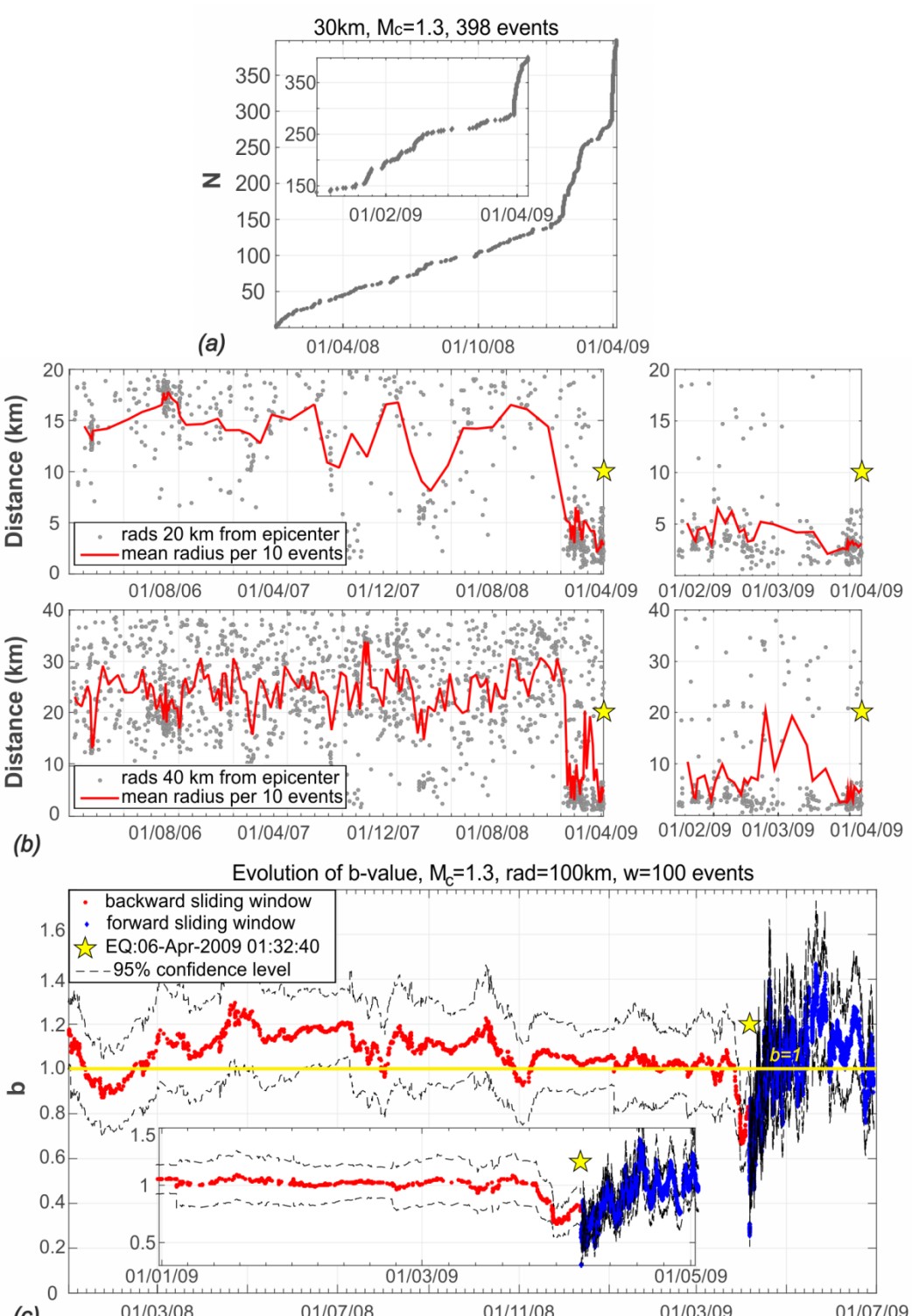

**Figure 2**. Time-space-size evolution of the L' Aquila foreshock sequence based on the INGV earthquake catalogue. (a) Cumulative number of earthquake events within a circle of radius of 30 km from the mainshock epicenter. The dramatic increase of the seismicity rate in about the last 3 months and particularly in the last 10 days (significance level 99%) before the mainshock of 6 April 2009 is evident. (b) Time evolution of the mean distance (red line) of foreshock epicenters from the mainshock epicenter (yellow star) for radius of 40 km (upper panel) and 20 km (lower panel). Calculation has been made for sequential but not overlapping sets of 10 events. Yellow star indicates the mainshock origin time. The mean distance gradually decreased being minimal in the last 10 days, that is during the strong foreshock stage. The right panels of the figure represent a zoomed area of the left ones showing the time evolution of the mean distance of foreshock epicenters from the mainshock epicenter for the last 2 months(c) Time evolution of the parameter $b$ calculated backwards before the L' Aquila mainshock (red) and forwards after the mainshock (blue). The parameter $b$ was calculated for sequential segments of the catalogue with a constant number of n=100 events and step of 1 event under the condition that the magnitude range in each segment was at least 1.4 as suggested by Papazachos (1974). The aim was to achieve stability in the results. If this condition was not satisfied then n was increased gradually with step of 1 event until the condition was satisfied. Black lines represent $\pm 2\sigma$ confidence intervals. Yellow star as in Fig. 2b. The parameter $b$ dropped gradually and reached values less than 0.70 in the last 10 days before the mainshock. In the aftershock period the $b$-value increased rapidly above 1. Yellow line in $b$=1makes the differences for different time windows more obvious.

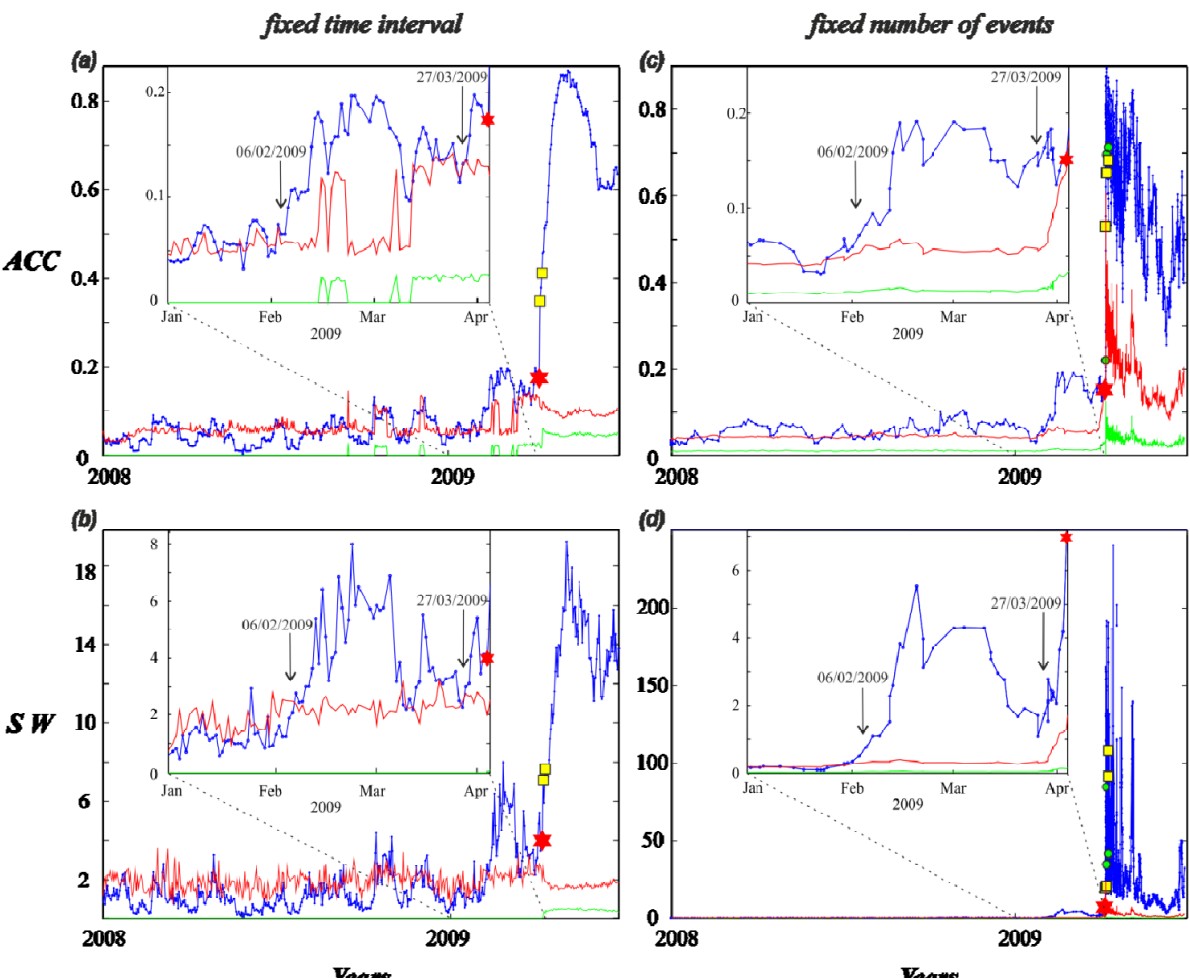

**Figure 3**. *ACC* and *SW* indices global statistical network measures from 01.01.2008 to 30.06.2009. Computations were performed in the space-window centered at the mainshock epicenter (42.42°N, 13.39°E), with radius $R_1 = 1^o$. Left column depicts the results obtained with a constant time shift of 1 day. Right column depicts the results obtained with a constant number of events shift. Earthquakes in the magnitude ranges of $4.3 \leq M \leq 4.9$ and $5.0 \leq M \leq 5.6$ are marked with green circles and yellow squares, respectively. The mainshock that occurred on April 6th, 2009 is marked with a red star. Red and green lines represent statistically significant levels of 95% and 5%, respectively (see description in the methods section).The inset shows a zoom in the period 1st January- 6th April, 2009. The arrow in the inset marks the onset of the statistically significant changes from the random network topology.

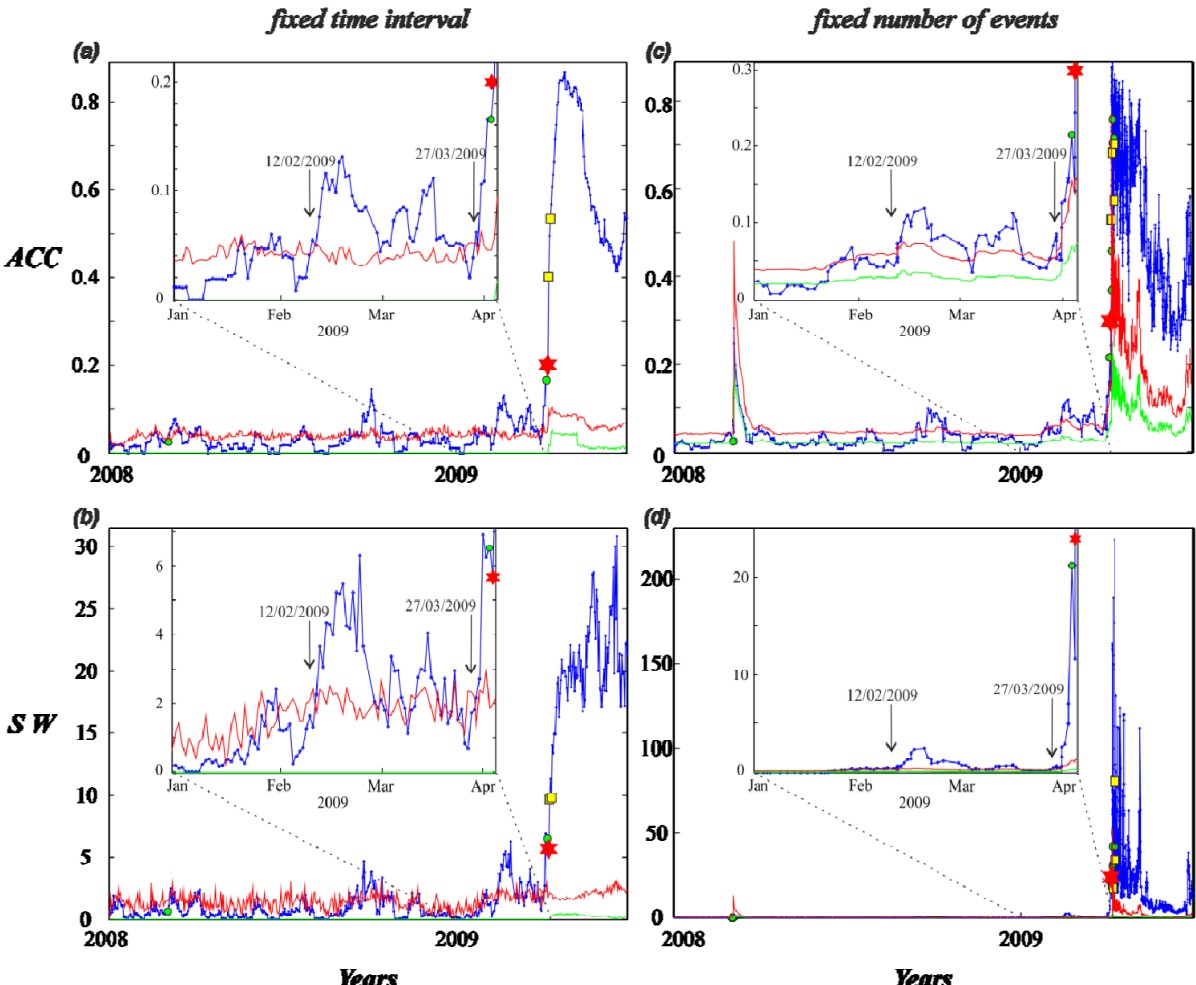

**Figure 4**. As in Fig. 3 for computations performed with radius $R_2 = 3^o$.

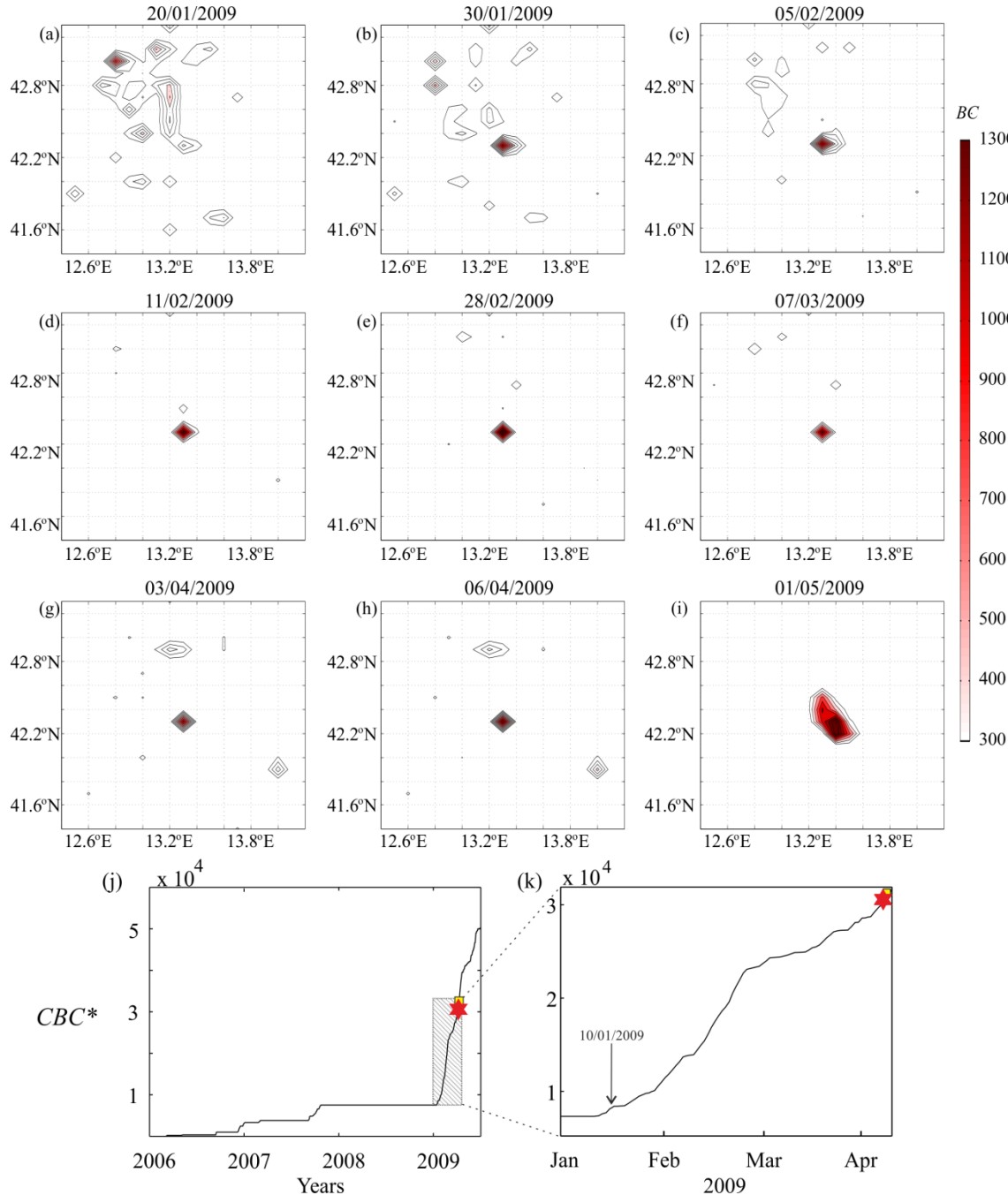

**Figure 5**. Betweeness centrality ($BC$) computations in the space-window centered at the mainshock epicenter (42.42°N, 13.39°E) with radius $R_1 = 1°$: (a-i) snapshots of the $BC$ maps for each cell from 20.01.2009 to 01.05.2009; (j, k) cumulative $BC$ computed at the mainshock cell ($CBC^*$) from 01.01.2006 to 30.06.2009 (j), and from 01.01.2009 to 06.04.2009 (k). The mainshock occurred on April 6th, 2009 (red star).

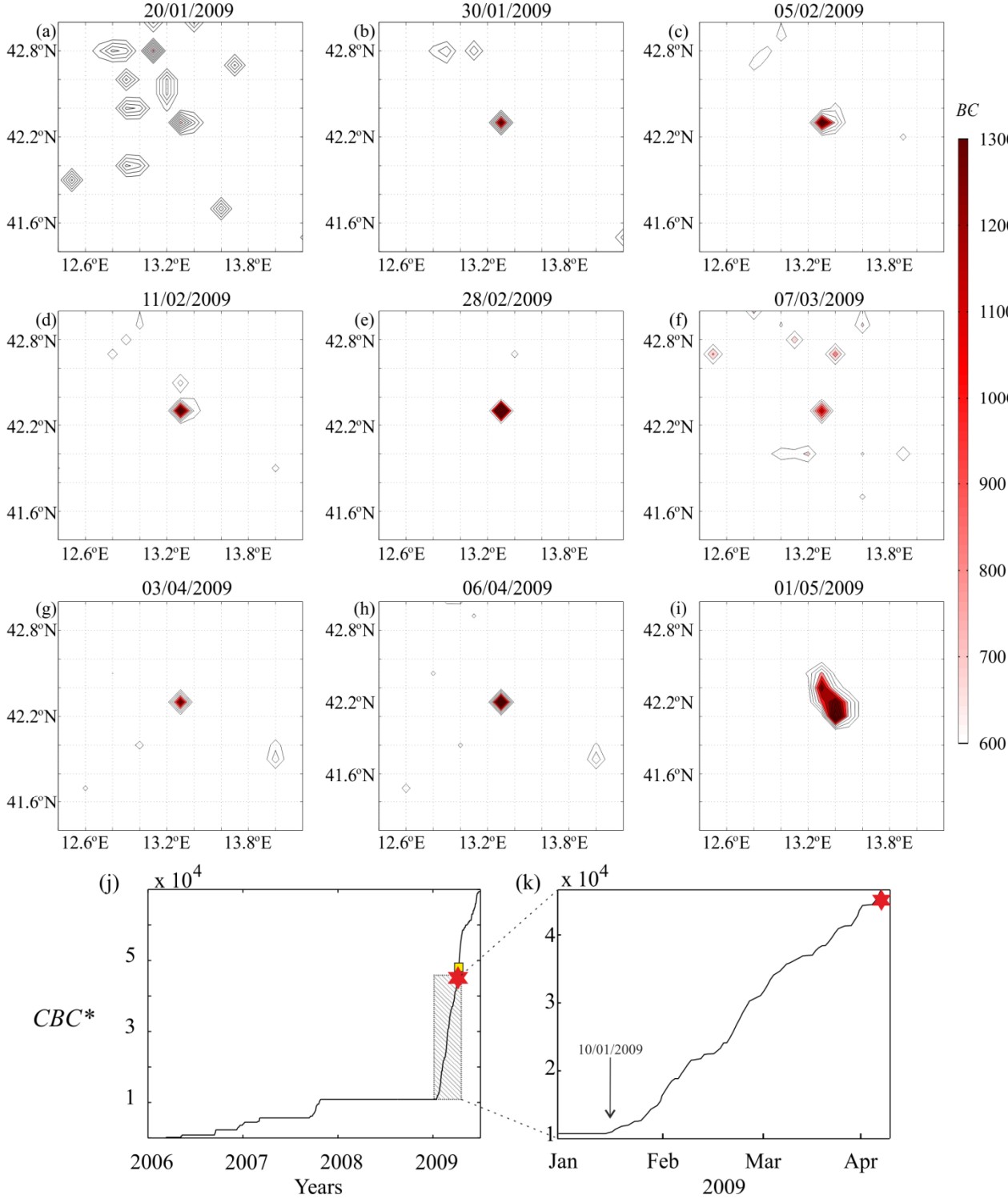

**Figure 6**. As in Fig. 5 for space-window with radius $R_2 = 3^o$.

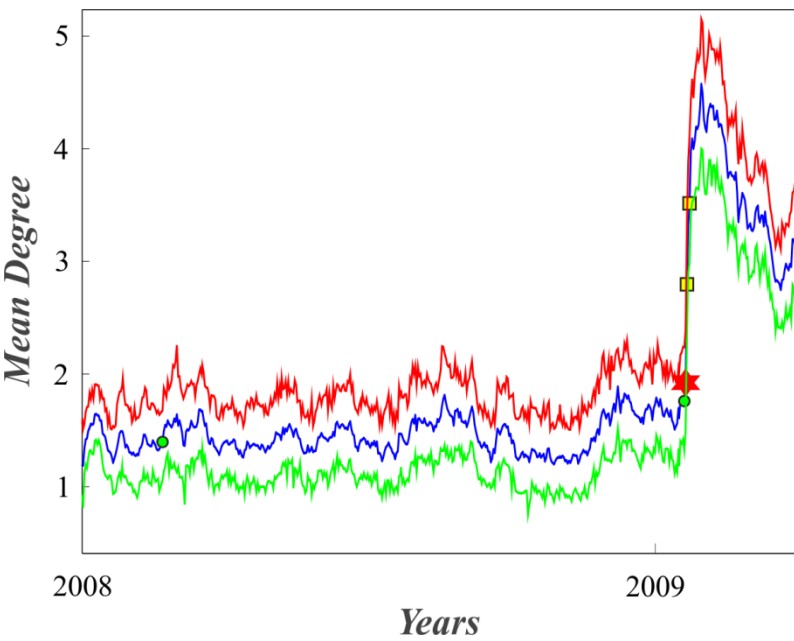

**Figure 7**. The mean degree of the seismic network (blue line) as computed using sliding windows with the constant time shift of 1 day. Red and green lines represent statistically significant levels of 95% and 5%, respectively, as obtained from an ensemble of equivalent random networks, (see description in the methods section).

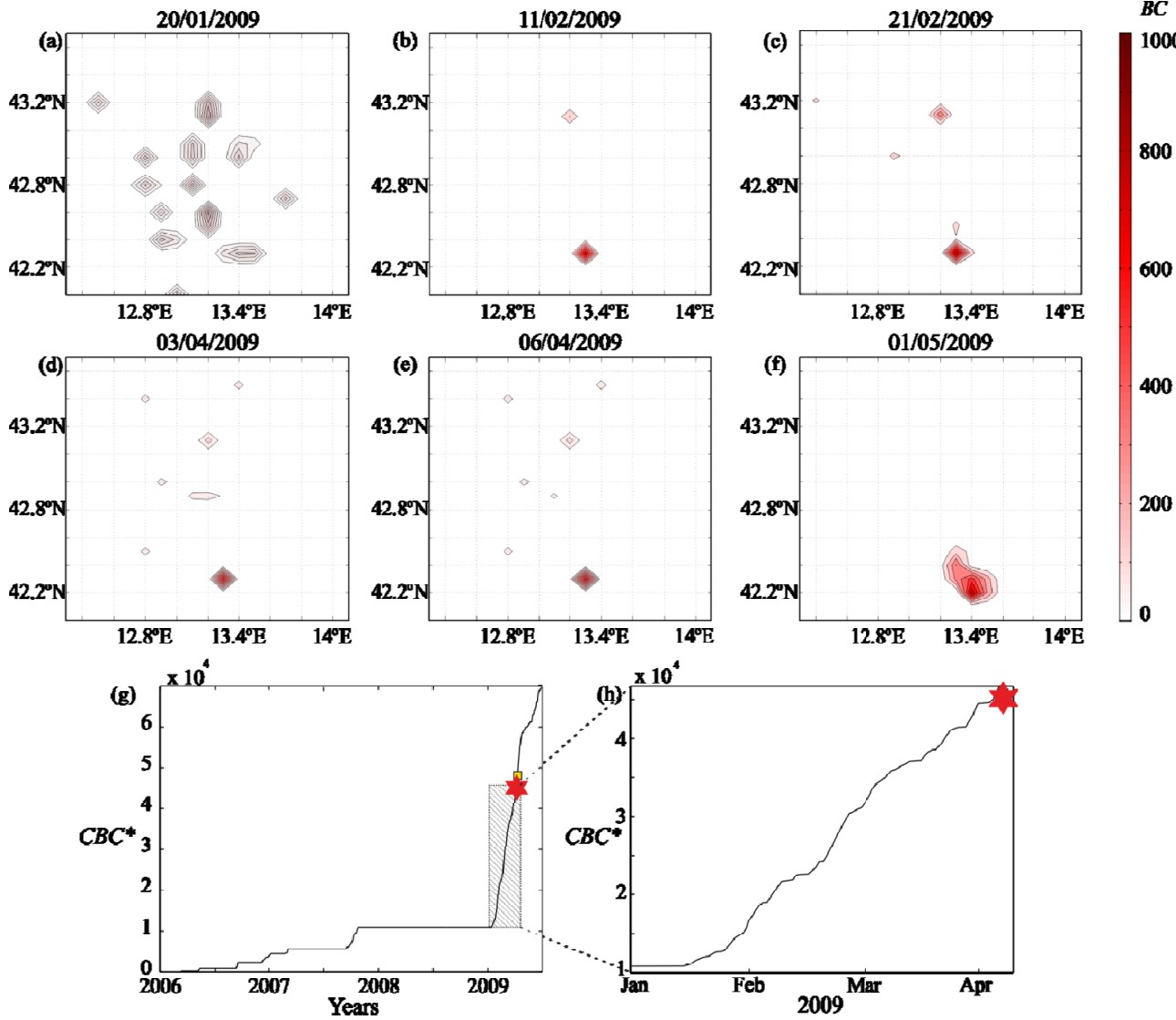

**Figure 8.** Betweeness centrality ($BC$) computations in the space-window centered at 42.00°N, 13.30°E (off the center of the epicenter) with radius $R_1 = 1^o$. The mainshock occurred on April 6[th], 2009 (red star): (a-i) snapshots of the $BC$ maps for each cell from 20.01.2009 to 01.05.2009; (a-e) $BC$ maps before the mainshock. (f) Cumulative $BC$ computed at the mainshock cell ($CBC^*$) from 01.01.2006 to 30.06.2009 (g), from 01.01.2009 to 06.04.2009 (h).

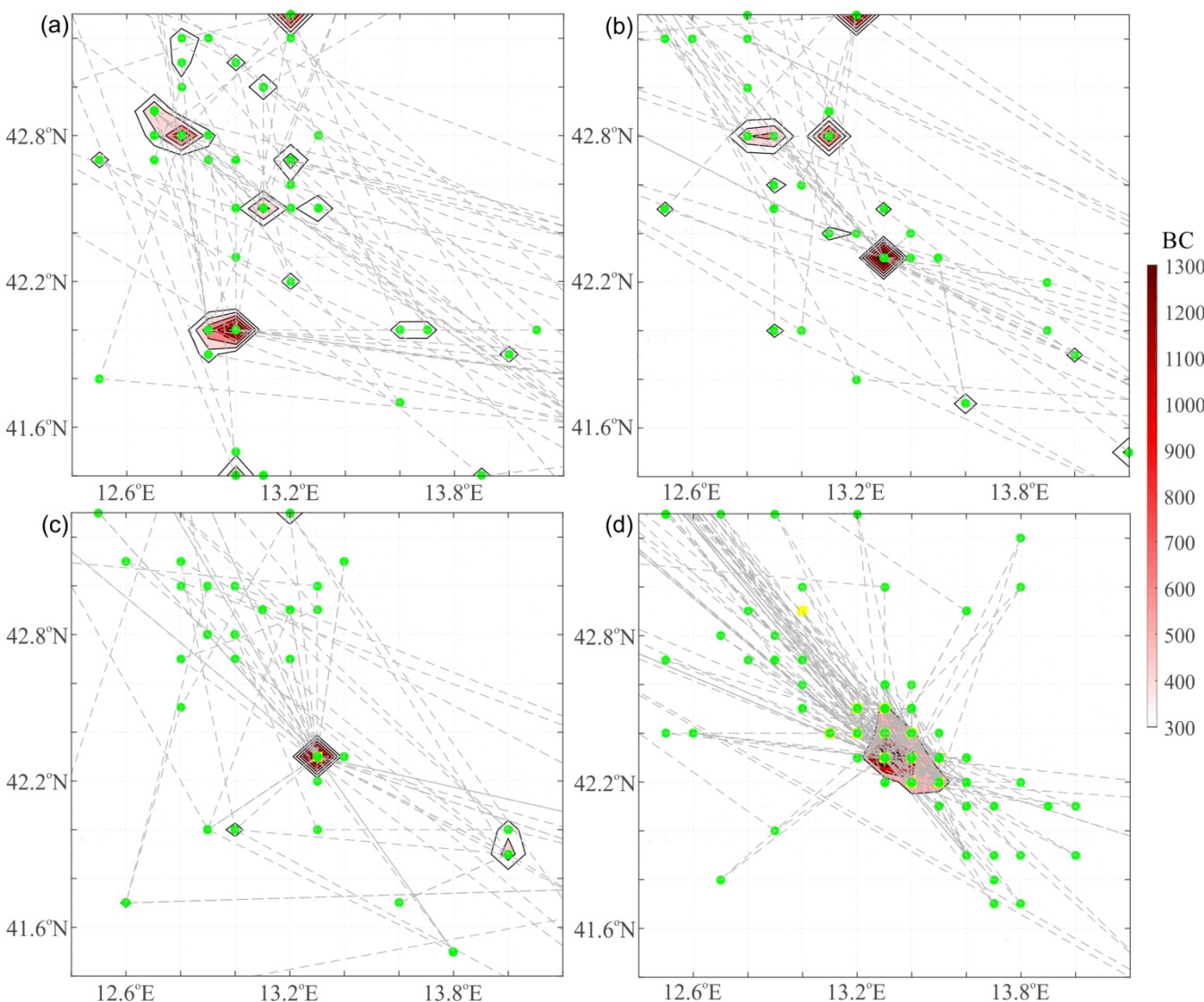

**Figure 9.** Snapshots of seismic networks overlaid on the betweenness centrality (BC) contours for $R_1 = 3°$ around the epicenter; green squares corresponds to events of $M \leq 3.0$ while yellow squares corresponds to events of $3.1 \leq M \leq 5$. Dash lines are the edges of the network that connect successive seismic events. The time window for each network is 15 days. (a) From 26 September until 10 October 2008, (b) From 14 to 27 January 2009, (c) From 20 March to 4 April 2009 and (d) From 1 to 16 April 2009.