# Peer review of "Foreshocks and Short-Term Hazard Assessment to Large Earthquakes using Complex Networks: the Case of the 2009 L'Aquila Earthquake"

_Nonlinear Processes in Geophysics, 2015_

## Referee Comment (RC1) · R. V. Donner (Referee) · 7 Mar 2016

In the last years, complex network theory has been applied in a variety of studies on the spatio-temporal organization of seismicity. The present manuscript by Daskalaki et al. adds to this increasing body of literature by reporting efforts to utilize network approaches for characterizing the foreshock activity associated with the 2009 L'Aquila earthquake. Similar evolving network analyses are still quite rare in the natural hazards community, and the present work could serve as an interesting case study demonstrating the potentials of this approach. However, in its present shape, the manuscript by

[Figure]

Daskalaki et al. leaves some questions open, which need to be addressed in a careful revision before this work can be recommended for final publication.

Major comments:

1. From the description of the utilized approach for earthquake network construction, it is not clear if the resulting network is considered directed or undirected. Since the construction is based on a temporal succession of events in some well-defined direction, a directed network representation appears reasonable. However, in such case, the definition of ACC would not be unique, since different motifs of three nodes would be accounted for. This aspect should be clarified.

2. The information provided by the evolving earthquake network analysis in terms of ACC and small-world index can hardly be interpreted without knowledge of the associated link density (or, alternatively, mean degree) and its variation with time. This information needs to be added. Notably, the path length of a network shows an ultimate relation with the link density, which would be reflected in the small-world index. A similar statement applies to the ACC: if we have a sparse network with low mean degree, the fraction of nodes with degree smaller than 2 can be expected to be larger than for networks with more edges. Such nodes contribute with a zero term to the calculation of the ACC. Hence, the temporal signatures of ACC reported by the authors could also trivially reflect different link densities during different time windows. A way to circumvent this problem would be replacing the ACC by the "network transitivity" or clustering coefficient as defined by Barrat and Weigt, Eur. Phys. J. B, 2000. A comparative discussion of both measures in terms of evolving networks can be found in Radebach et al., Phys. Rev. E, 2013.

3. The authors relate the "more clustered seismicity pattern" identified by ACC to "the emergence of few nodes with higher centrality [supposedly betweenness centrality?], which act as hubs" (p.6, ll.5-6). This is not clear, which can already be seen from the previous comment.

4. It is not clear why network measures are necessary to identify the strong spatial clustering prior to the L'Aquila mainshock. Couldn't standard methods of spatial statistics serve the same purpose?

5. The authors claim that "the topological measures appear to outperform other observables reported in previous statistical work" (p.7, ll.23-24) without clarifying which previous variables are meant. No corresponding references are given, nor does the manuscript contain a detailed comparative study for the considered foreshock sequence. Since the performance statement is repeated twice on p.8, the least to be expected is further detailed information on this aspect. I also don't think that a comparative performance assessment is possible based on just a single case study like the one reported here.

Minor comments:

6. On p.3, l.11, I would speak of "hindcasting" rather than "forecasting", since the corresponding analysis has been made a posteriori after the event occurred.

7. In order to better understand the meaning of the parameters b and r, please give the Gutenberg-Richter relationship explicitly in the text.

8. In what sense has the foreshock sequence been produced by a physical process "dominated by a strong chaotic component" (p.3, l.24)?

9. On p.4, ll.25-28, the symbols $N_a$ etc. are used to denote event indices, but rather resemble window sizes. Using different symbols might help avoiding possible confusion with standard notions of other papers.

10. The motivation for the selection of ACC and small-world index is not clear. Instead of the small-world index (which should be accompanied by the original reference), it would make more sense to study ACC and APL, since both are independent while ACC and small-world index are not.

11. The statement that the network properties are obtained "by averaging the corresponding properties over all the nodes of the network" (p.5, l.1) is not quite obvious for the small-world index.

12. The term "degree" should be briefly explained at its first appearance in the text. Network scientists know this term very well, but this is not necessarily the case for seismologists.

13. The term "random regular graph" (p.5, l.30) does not exist – you have either a random or a regular graph.

14. Can one motivate the idea that "hubs that could serve as potential epicenter locators" (p.6, ll.14-15) from known seismological principles?

15. The authors state that "the recognition of the seismicity anomaly by topological measures does not discriminate the seismicity style, i.e., foreshocks, swarms or aftershocks" (p.8, ll.18-19). This is surely correct for the present network construction approach. In turn, other recent types of construction mechanisms have been used for declustering earthquake catalogs and, thus, identifying fore- and aftershock sequences (Jimenez et al., EPJST, 2009). It appears reasonable to add a corresponding comment.

16. For the definition of the small-world effect in complex networks, both ACC and APL are commonly taken into account together. What the authors report on p.9, l.17, for the behavior of APL seems not to fully comply with the common view. I recommend consulting the seminal work by Watts and Strogatz (Nature, 1998) for details.

17. It is not clear what the authors mean by "local [network] property" (p.9, l.24). BC is clearly a node property, however, its computation requires global linkage information on the entire network. In this regard, the term "local property" might be misleading.

18. p.9, ll.24-25: "BC(i) indicates that the i[-th] node acts as a central node influencing most of the other nodes" – how has this influence to be understood in the context of the considered earthquake networks?

19. In Fig. 2c, an additional horizontal line at b=1 might help visualizing the differences

for different time windows as discussed by the authors. In the caption, the right panels of Fig. 2b should be mentioned (even though they only represent a zoom of parts of the right panels).

20. The cumulated BC (CBC*) is not clearly defined in the text, and its definition and meaning are not clear from the text. This aspect needs to be clarified.

21. I recommend including Fig. S1 in the main text. A supplementary information for just a single figure does not seem necessary.

22. Throughout the manuscript, there seem to be numerous (yet minor) language issues like confusion of singular and plural forms, use of articles, word order, etc. Careful proofreading is recommended prior to resubmission.

Despite this number of comments, I think that the present work is interesting and might provide relevant insights into the spatio-temporal organization of seismicity preceding the 2009 L'Aquila earthquake. I am not yet convinced that similar observations also apply to other major earthquakes, but corresponding follow-up studies appear reasonable given the promising results of the present work. In this regard, I would warmly welcome a revised version of this manuscript that has addressed all points raised above.

---

## Referee Comment (RC2) · A. De Santis (Referee) · 27 Mar 2016

**Comments to "Foreshocks and short-term hazard assessment to large earthquakes using complex networks: the case of the 2009 L'Aquila earthquake" by Daskalaki et al. submitted to NPG**

A. De Santis

The paper is an interesting application of a statistical network analysis to the 2009 L'Aquila seismic sequence. My indication is for a major revision: I have some specific requests that once are met properly the paper might be published.  They are indicated in the due order in the following.

**Major point**

As I understand, the network analysis does not take into account the different magnitudes of the foreshocks. This means that it could depend on the chosen minimum magnitude of the dataset (now Mc=1.3). I think this is the weakest point of this kind of analysis. Results from some other different choices should be shown, in order to prove that the choice is not critical.

**Secondary points**

- I suppose the selection of the seismic events is limited by the depth so considering only shallow earthquakes, but this is not said. Could you please specify it? And, how much critical is the choice (e.g. showing results for two other depth choices)?

- I believe that some passages of the manuscript would be strengthen by adding some proper references that are now missing. I can suggest some (but the Authors can add alternatives), as the following:
Pag.2 line 8. After "(e.g. Bufe and Varnes, 1993)" I would add the sentence: "A recent revision of the method has been proposed in order to cope with some previous limitations (De Santis et al., 2015)."
Pag.3 line 14. I would add at the end of the present sentence the following: "However, around a year before the mainshock possible effects due to fluid migration was found from magnetic data analyses (Cianchini et al., 2012)."

Pag. 7 line 23. After "its occurrence" I would insert the following: "(look also De Santis et al. 2015)"
Pag.8 line 1. Please, after "2010" insert: "; De Santis et al., 2011".

Fig.2 caption, pag. 16 line 13. When you write "$2\sigma$ confidence intervals" do you really mean "$\pm 2\sigma$ confidence intervals"  or "$\pm\sigma$ confidence intervals"? Please clarify.

Fig. 5 pag. 19 and Fig. 6 pag. 20. Although you already provide the spatial distribution of the earthquakes in Fig. 1, I believe that these Figures would be clearer if associated with the progressive spatial distribution of earthquakes in each frame, even provided in a separate Figure (if the points decrease clarity in reading).

**Minor points**
- Pag.2 line 2. I would insert "generally" between "foreshocks" and "increases".
-Pag. 3 line25. Please correct "2011" with "2010".
- Pag 4 line19.  Please correct "0.1o" with "0.1°"
- Page 5 line14. Please correct "km3" with "$km^3$"
- Watts and Strogatz, 1998 (indicated in the Appendix at pag. 8, line 28-29) is missing in the list of references.
- Pag. 10 lines 5-9. Amoruso et al. 2010 and Albert and Barabasi 2002 should be put in reverse order in the reference list.

**References (not already present in the manuscript)**

Cianchini G., A. De Santis, D. R. Barraclough, L. X. Wu, and K. Qin, 2012. Magnetic transfer function entropy and the 2009 Mw = 6.3 L'Aquila earthquake (Central Italy), *Nonlin. Processes Geophys.*, 19, pp. 401-409, doi:10.5194/npg-19-401-2012.

De Santis A., Cianchini G., Di Giovambattista, 2015. Accelerating moment release revisited: examples of application to Italian seismic sequences, *Tectonophysics*, 639, 82-98, 10.1016/j.tecto.2014.11.015.

---

## Author Comment (AC1) · 7 May 2016

We would like to thank the referees for taking the time and effort to review our manuscript and for their positive, constructive and indeed thorough comments. In the revised version of our manuscript we will accommodate all of their remarks.

1st reviewer

1. From the description of the utilized approach for earthquake network construction, it is not clear if the resulting network is considered directed or undirected. Since the construction is based on a temporal succession of events in some well-defined direction, a directed network representation appears reasonable. However, in such case, the definition of ACC would not be unique, since different motifs of three nodes would be accounted for. This aspect should be clarified.

Response. The proposed approach results to a directed network. In the appendix we give the definitions of statistical measures for directed networks. For example the first equation in the appendix gives the clustering coefficient for a directed graph, as it is defined by Fagiolo, Physical Review E 76, 026107 2007. According to this definition, the clustering coefficient is again the ratio between all directed triangles actually formed by node i and the number off all possible triangles that node i could form. This definition gives a natural extension of binary undirected networks. Finally in the case of random graphs the expected mean ACC is p (same as the random undirected networks) where p is the probability that two nodes will be connected. However reading again the manuscript based on the comment of the reviewer indeed this is not so clear written. In the revised version we intend to clarify better this issue. We will also add in, the relevant references for the definition of the ACC for directed networks.

2. The information provided by the evolving earthquake network analysis in terms of ACC and small-world index can hardly be interpreted without knowledge of the associated link density (or, alternatively, mean degree) and its variation with time. This information needs to be added. Notably, the path length of a network shows an ultimate relation with the link density, which would be reflected in the small-world index. A similar statement applies to the ACC: if we have a sparse network with low mean degree, the fraction of nodes with degree smaller than 2 can be expected to be larger than for networks with more edges. Such nodes contribute with a zero term to the calculation of the ACC. Hence, the temporal signatures of ACC reported by the authors could also trivially reflect different link densities during different time windows. A way to circumvent this problem would be replacing the ACC by the "network transitivity" or clustering coefficient as defined by Barrat and Weigt, Eur. Phys. J. B, 2000. A comparative discussion of both measures in terms of evolving networks can be found in

Radebach et al., Phys. Rev. E, 2013.

Response. We agree with the reviewer that there is a clear dependency between the link density and the corresponding network measures. For that reason in the manuscript we present also the "fixed number of events"-way of constructing the underlying networks. This fixes the link density to a constant and in this way the effect of the link density is factored out (see e.g. Figure 3, right column). We calculated the mean degree (not shown) in order to perform statistical significance analysis (page 5 line 5-7). Within the period before the main seismic event, we found no significant statistical differences with respect to the mean degree. However we agree with the reviewer that an extra information of the link density is required and we will do so in the revised manuscript.

3. The authors relate the "more clustered seismicity pattern" identified by ACC to "the emergence of few nodes with higher centrality [supposedly betweenness centrality?], which act as hubs" (p.6, ll.5-6). This is not clear, which can already be seen from the previous comment.

Response. We agree with the reviewer that this is not so clear. We will clarify better in the revised manuscript taking also into account the previous comment.

4. It is not clear why network measures are necessary to identify the strong spatial clustering prior to the L'Aquila mainshock. Couldn't standard methods of spatial statistics serve the same purpose?

Response. Certainly standard methods of spatial statistics may serve the same target and have been done in previous papers which we cite in our manuscript in section 2. What we propose here is an alternative/complementary way that may facilitate and strengthen our arsenal in accessing strong spatial clustering and identifying "tip" points and particular distinct spatio-temporal patterns in the behavior of the seismicity. One of the relative advantages that comes from complex network theory is that networks may be used to identify efficiently, within the nonlinear dynamics theory framework, phasetransitions that mark the onset of big changes in the underlying seismicity. Actually, as we mention in the manuscript, this line of research is motivated by the concept of self-organized criticality (Bak, 1996) which models structural phase transitions from random to scale-free spatial correlations between seismic events (e.g. Sammis and Sornette, 2002). Yet, the generalization and reliability of the outcomes of this relatively new approach remains an open question. The wealth of statistical measures of the reconstructed network activity (such as the small-worldness, path-length, local and global clustering coefficient, betweenness centrality) offer many different views and tools for characterizing the underlying varying topology. For the particular case, the proposed approach looks promising as identified (retrospectively as all other methods until now) quite efficiently, about two months before the mainshock, the location of the mainshock epicenter. From this point of view one may recognize that the Betweenness centrality measure outperformed the classic space-time seismicity statistics that have been reported until now. In the revised version we intend to highlight more these issues in the Discussion Section.

5. The authors claim that "the topological measures appear to outperform other observables reported in previous statistical work" (p.7, ll.23-24) without clarifying which previous variables are meant. No corresponding references are given, nor does the manuscript contain a detailed comparative study for the considered foreshock sequence. Since the performance statement is repeated twice on p.8, the least to be expected is further detailed information on this aspect. I also don't think that a comparative performance assessment is possible based on just a single case study like the one reported here.

Response. In the manuscript we state the following: "Interestingly, in the a posteriori analysis of the 2009 L' Aquila seismic sequence, the betweeness centrality and its cumulative expression, which are local statistical network measures, started to pinpoint the nucleation area of the forthcoming strong earthquake two months before its occurrence. In this sense, the topological measures appear to outperform other observables

reported in previous statistical works in terms of the detection of the onset of "persistent" steep changes in the system's observables." Hence we refer to the BC and its Cumulative. In the revised version we will cite and perform a quantitative comparative analysis with other studies. We don't also claim that the proposed method will always outperform all other methods. As we state in the introduction ÂńYet, the generalization and reliability of the outcomes of this relatively new approach remains an open questionÂż. Indeed more studies are needed and our work contributes exactly to this open question, appearing as promising.

6. On p.3, l.11, I would speak of "hindcasting" rather than "forecasting", since the corresponding analysis has been made a posteriori after the event occurred.

Response. We agree with the reviewer, we intend to change this as suggested.

7. In order to better understand the meaning of the parameters b and r, please give the Gutenberg-Richter relationship explicitly in the text.

Response. Page 3, l. 18 . . ...where b is the slope of the magnitude-log frequency or G-R relation: $\log N = a - bM$ (1) where N is the cumulative number of events of magnitude equal to or larger than M and a, b are parameters determined by the data.

8. In what sense has the foreshock sequence been produced by a physical process "dominated by a strong chaotic component" (p.3, l.24)?

Response. According to the finding of De Santis et al. (2010) there was a strong chaotic component driven by the accelerated seismic strain release. This is a result of De Santis et al. (2010), not ours. Maybe this is not clearly said. We will write it in a clear manner in the revised text.

9. On p.4, ll.25-28, the symbols Na etc. are used to denote event indices, but rather resemble window sizes. Using different symbols might help avoiding possible confusion with standard notions of other papers.

Response. We will change symbols as suggested.

10. The motivation for the selection of ACC and small-world index is not clear. Instead of the small-world index (which should be accompanied by the original reference), it would make more sense to study ACC and APL, since both are independent while ACC and small-world index are not.

Response. The small-world index is an important statistical measure that reveals how a network structure deviates from random ones that account for regular seismicity. Thus, it can be used to characterize phase transitions that mark the onset of relatively big changes in the underlying topology. The original definition comes from Humphries et al. 2006 and gives an efficient way of characterizing all together the structure of the network with respect to the relation between clustering coefficient and path length.

11. The statement that the network properties are obtained "by averaging the corresponding properties over all the nodes of the network" (p.5, l.1) is not quite obvious for the small-world index.

Response. This statement is not for the small-world index but for the path lengths and clustering. We will clarify this in the revised version of the manuscript.

12. The term "degree" should be briefly explained at its first appearance in the text. Network scientists know this term very well, but this is not necessarily the case for seismologists.

Response.We agree with the reviewer. We will define it in the revised version.

13. The term "random regular graph" (p.5, l.30) does not exist – you have either a random or a regular graph.

Response. Allows us to say that the term "random regular graph" exists and refers to random graphs with uniform degree (see e.g. https://en.wikipedia.org/wiki/Random_regular_graph). However we agree that it is not a common terminology to seismologists. We will clarify this in the revised version.

14. Can one motivate the idea that "hubs that could serve as potential epicenter loca­tors" (p.6, ll.14-15) from known seismological principles?

Response. p.6 l. 24 ….node related to the mainshock epicenter. This observation along with the drop of the b-value (e.g. Papadopoulos et al., 2010) indicates stress increase close to the mainshock epicenter, thus underlying the physical link between the centralization of the ðİŘţC distribution and the mainshock nucleation process. The ðİŘţC of all other nodes do not change their values significantly as we present in the manuscript.

15. The authors state that "the recognition of the seismicity anomaly by topological measures does not discriminate the seismicity style, i.e., foreshocks, swarms or af­tershocks" (p.8, ll.18-19). This is surely correct for the present network construction approach. In turn, other recent types of construction mechanisms have been used for declustering earthquake catalogs and, thus, identifying fore- and aftershock sequences (Jimenez et al., EPJST, 2009). It appears reasonable to add a corresponding comment.

Response. The recognition of the seismicity anomaly by topological measures, how­ever, does not discriminate the seismicity style, i.e. foreshocks, swarms or aftershocks. The foreshock nature of the anomaly was detected only by the a posteriori knowl­edge that the earthquake sequence concluded with a mainshock. In seismology sev­eral branching type models have been used to identify space-time clusters, e.g. the epidemic-type aftershock sequence (ETAS) model (OGATA, 1998). Although it was tested for analyzing clustering features of foreshocks (e.g. ZHUANG and OGATA, 2006) no standard method has been introduced so far for the foreshock recognition beforehand. This is also valid for the several techniques introduced for declustering earthquake catalogues (e.g. Jimenez et al., EPJST, 2009). However, the classic seis­micity statistics yields possibilities for such a discrimination beforehand thanks to the discriminatory power of the b parameter which drops significantly during foreshocks. The drop of b is not only of statistical but also of geophysical value. Therefore, investi­gating for an equivalent discriminatory topological measure is a challenge. We will add

a corresponding comment as suggested.

16. For the definition of the small-world effect in complex networks, both ACC and APL are commonly taken into account together. What the authors report on p.9, l.17, for the behavior of APL seems not to fully comply with the common view. I recommend consulting the seminal work by Watts and Strogatz (Nature, 1998) for details.

Response. The small-world index was first introduced by Humphries et al. 2006 http://www.ncbi.nlm.nih.gov/pmc/articles/PMC1560205/ and we use the same definition as appears in that paper. In line 17 we say that the APL for small world network is of the same order as the equivalent random structure. This is in line with the definition and properties of the APL (see Watts, 1999). http://www.cc.gatech.edu/~mihail/D.8802readings/watts-swp.pdf). Actually pure random graphs exhibit relatively small average-path lengths. However, indeed this is not so clear we will clarify this issue and cite the appropriate papers.

17. It is not clear what the authors mean by "local [network] property" (p.9, l.24). BC is clearly a node property, however, its computation requires global linkage information on the entire network. In this regard, the term "local property" might be misleading.

Response. We agree that could be misdealing but this is the standard terminology used in the field. We will clarify better in the revised version.

18. p.9, ll.24-25: "BC(i) indicates that the i[-th] node acts as a central node influencing most of the other nodes" – how has this influence to be understood in the context of the considered earthquake networks?

Response. This means that the seismicity revolves around this particular point. We will explain better in the revised version.

19. In Fig. 2c, an additional horizontal line at b=1 might help visualizing the differences for different time windows as discussed by the authors. In the caption, the right panels of Fig. 2b should be mentioned (even though they only represent a zoom of parts of he

right panels).

Response. We will make these changes as suggested.

20. The cumulated BC (CBC*) is not clearly defined in the text, and its definition and meaning are not clear from the text. This aspect needs to be clarified.

Response. We will describe it better in a clear manner in the revised version.

21. I recommend including Fig. S1 in the main text. A supplementary information for just a single figure does not seem necessary.

Response. We agree and we will include it in the main text.

22. Throughout the manuscript, there seem to be numerous (yet minor) language issues like confusion of singular and plural forms, use of articles, word order, etc. Careful proofreading is recommended prior to resubmission.

Response. We will correct all these issues.

2nd reviewer 1. Major point As I understand, the network analysis does not take into account the different magnitudes of the foreshocks. This means that it could depend on the chosen minimum magnitude of the dataset (now Mc=1.3). I think this is the weakest point of this kind of analysis. Results from some other different choices should be shown, in order to prove that the choice is not critical.

Response. Îďhe selection of the minimum magnitude of the data set was not done ad-hoc. The earthquake catalogue was tested for data completeness on the basis of the G-R diagram as done in the classical statistics. This is an important task as mishandling data may subsequently lead to wrong evaluations for a series of important features, such as seismic rate changes, with implications for the identification of earthquake sequences, e.g. aftershocks. Indeed the above is not highlighted enough in the text. We will explain it better in the revised version.

2. Secondary points - I suppose the selection of the seismic events is limited by the

depth so considering only shallow earthquakes, but this is not said. Could you please specify it? And, how much critical is the choice (e.g. showing results for two other depth choices)?

Response. We have considered the earthquake catalogue of the area without threshold in the focal depth. However, the majority (nearly 97%) of the events have focal depth less than 30 km, which means that they are shallow. The remaining may have depth up to about 45 km but we did not remove them from the catalogue allowing for some error to be involved in the depth determination. As a consequence, the data set that we used practically represents the shallow seismogenic layer. We will write it explicitly in the revised version.

- I believe that some passages of the manuscript would be strengthen by adding some proper references that are now missing. I can suggest some (but the Authors can add alternatives), as the following: Pag.2 line 8. After "(e.g. Bufe and Varnes, 1993)" I would add the sentence: "A recent revision of the method has been proposed in order to cope with some previous limitations (De Santis et al., 2015)." Pag.3 line 14. I would add at the end of the present sentence the following: "However, around a year before the mainshock possible effects due to fluid migration was found from magnetic data analyses (Cianchini et al., 2012)."

Response. We agree and we will add the appropriate references in the revised manuscript.

Pag. 7 line 23. After "its occurrence" I would insert the following: "(look also De Santis et al. 2015)".

Response. We agree. We will make the necessary change as suggested.

Pag.8 line 1. Please, after "2010" insert: "; De Santis et al., 2011".

Response. We agree and we will do so.

Fig.2 caption, pag. 16 line 13. When you write "2$\sigma$ confidence intervals" do you really

mean "$\pm 2\sigma$ confidence intervals" or "$\pm\sigma$ confidence intervals"? Please clarify.

Response. We mean "$\pm 2\sigma$ confidence intervals. We will clarify this in the revised version.

-Fig. 5 pag. 19 and Fig. 6 pag. 20. Although you already provide the spatial distribution of the earthquakes in Fig. 1, I believe that these Figures would be clearer if associated with the progressive spatial distribution of earthquakes in each frame, even provided in a separate Figure (if the points decrease clarity in reading).

Response. We will try to do that.

Minor points

Response. We agree and we will fix all of them in the revised version.

References

Response. We will add the suggested references

Cianchini G., A. De Santis, D. R. Barraclough, L. X. Wu, and K. Qin, 2012. Magnetic transfer function entropy and the 2009 Mw = 6.3 L'Aquila earthquake (Central Italy), Nonlin. Processes Geophys., 19, pp. 401-409, doi:10.5194/npg-19-401-2012.

De Santis A., Cianchini G., Di Giovambattista, 2015. Accelerating moment release revisited: examples of application to Italian seismic sequences, Tectonophysics, 639, 82-98, 10.1016/j.tecto.2014.11.015.

---

## Author Response (AR1)

We would like to thank the referees for taking the time and effort to review our manuscript andfor their positive, constructive and thorough comments. In the revised version of our manuscript we have accommodate all of their remarks.

**1st reviewer**

1. From the description of the utilized approach for earthquake network construction,it is not clear if the resulting network is considered directed or undirected. Since theconstruction is based on a temporal succession of events in some well-defined direction,a directed network representation appears reasonable. However, in such case,the definition of ACC would not be unique, since different motifs of three nodes wouldbe accounted for. This aspect should be clarified.

**Response**. The proposed approach results to a directed network. In the appendix we give the definitions of statistical measures for directed networks. For example the first equation in the appendix gives the clustering coefficient for a directed graph, as it is defined byFagiolo, Physical Review E 76, 026107 2007. According to this definition, the clustering coefficient is again the ratio between all directed triangles actually formed by nodeiand the number off all possible triangles that node i could form. This definition gives a natural extension of binary undirected networks. Finally in the case of random graphs the expected mean ACC is p (same as the random undirected networks) where p is the probability that two nodes will be connected.

However reading again the manuscript based on the comment of the reviewer indeed this is not so clearly written. In the revised version we intend to clarify better this issue. We will also add the relevant references for the definition of the ACC for directed networks.

**Change**.Pag. 5, Lines8-16 and appendix pag.11 lines 14-17.

2. The information provided by the evolving earthquake network analysis in terms ofACC and small-world index can hardly be interpreted without knowledge of the associatedlink density (or, alternatively, mean degree) and its variation with time. Thisinformation needs to be added. Notably, the path length of a network shows an ultimaterelation with the link density, which would be reflected in the small-world index.A similar statement applies to the ACC: if we have a sparse network with low meandegree, the fraction of nodes with degree smaller than 2 can be expected to be largerthan for networks with more edges. Such nodes contribute with a zero term to the calculationof the ACC. Hence, the temporal signatures of ACC reported by the authorscould also trivially reflect different link densities during different time windows. A wayto circumvent this problem would be replacing the ACC by the "network transitivity" orclustering coefficient as defined by Barrat and Weigt, Eur. Phys. J. B, 2000. A comparativediscussion of both measures in terms of evolving networks can be found inRadebach et al., Phys. Rev. E, 2013.

**Response**. We agree with the reviewer that there is a clear dependency between the link density and the corresponding network measures. For that reason in the manuscript we present also the "fixed number of events"-way of constructing the underlying networks. This fixes the link density to a constant and in this way the effect of the link density is factored out(see e.g. Figure 3, right column). We calculated the mean degree (not shown) in order to perform statistical significance analysis (page 5 line 5-7). Within the period before the main seismic event, we found no significant statistical differences with respect to the mean degree. However we agree with the reviewer that extra information of the link density is required and we will do so in the revised manuscript.

**Change**. Pag. 6, Lines 31-32, Pag.7, Lines 1-2 and also new figure 7.

3. The authors relate the "more clustered seismicity pattern" identified by ACC to "theemergence of few nodes with higher centrality [supposedly betweenness centrality?],which act as hubs" (p.6, ll.5-6). This is not clear, which can already be seen from theprevious comment.

**Response**.We agree with the reviewer that this is not so clear. We will clarify better in the revised manuscript taking also into account the previous comment.

**Change**.Pag. 6,Line. 21-25.

4. It is not clear why network measures are necessary to identify the strong spatialclusteringprior to the L'Aquila mainshock. Couldn't standard methods of spatial statisticsserve the same purpose?

**Response**. Certainly standard methods of spatial statistics may serve the same target and have been done in previous papers which we cite in our manuscript in section 2. What we propose here is an alternative/complementary way that may facilitate and strengthen our arsenal in accessing strong spatial clustering and identifying "tip" points and particular distinct spatio-temporal patterns in the behavior of the seismicity. One of the relative advantages that come from complex network theory is that networks may be used to identify efficiently, within the nonlinear dynamics theory framework, phase-transitions that mark the onset of big changes in the underlying seismicity. Actually, as we mention in the manuscript, this line of research is motivated by the concept of self-organized criticality (Bak, 1996) which models structural phase transitions from random to scale-free spatial correlations between seismic events (e.g. Sammis and Sornette, 2002). Yet, the generalization and reliability of the outcomes of this relatively new approach remains an open question.The wealth of statistical measures of the reconstructed network activity (such as the small-worldness, path-length, local and global clustering coefficient, betweenness centrality) offers many different views and tools for characterizing the underlying varying topology.

For the particular case, the proposed approach looks promising as identified (retrospectively as all other methods until now) quite efficiently, about two months before the mainshock, the location of the mainshock epicenter.

In the revised version we intend to highlight more these issues in the Discussion Section.

**Change**.Pag. 8-9, section 5. Discussion

5. The authors claim that "the topological measures appear to outperform other observablesreported in previous statistical work" (p.7, ll.23-24) without clarifying whichprevious variables are meant. No corresponding references are given, nor does themanuscript contain a detailed comparative study for the considered foreshock sequence.

Since the performance statement is repeated twice on p.8, the least to beexpected is further detailed information on this aspect. I also don't think that a comparativeperformance assessment is possible based on just a single case study like theone reported here.

**Response**. In the manuscript we included in the discussion section a review of the studies using other classical statistical measures. Now we have put out the word "outperform" and instead we write that our analysis provides an alternative way to describe the spatiotemporal precursory seismicity changes. In the introduction we also state that «Yet, the generalization and reliability of the outcomes of this relatively new approach remains an open question». Indeed more studies are needed and our work contributes exactly to this open question, appearing as promising.

**Change**.Pag. 8-9, section 5. Discussion

6. On p.3, l.11, I would speak of "hindcasting" rather than "forecasting", since thecorresponding analysis has been made a posteriori after the event occurred.

**Response**. We agree with the reviewer, we intend to change this as suggested.

**Change**. Pag.3, Line 11.

7. In order to better understand the meaning of the parameters b and r, please give theGutenberg-Richter relationship explicitly in the text.

**Response**. Page 3, l. 18 …..where b is the slope of the magnitude-log frequency or G-R relation:

log N = a – bM   (1)

where N is the cumulative number of events of magnitude equal to or larger than M and a, b are parameters determined by the data.

**Change**.Pag. 3, Line 20-24.

8. In what sense has the foreshock sequence been produced by a physical process"dominated by a strong chaotic component" (p.3, l.24)?

**Response**. According to the finding of De Santis et al. (2010) there was a strong chaotic component driven by the accelerated seismic strain release. This is a result of De Santis et al. (2010), not ours. Maybe this is not clearly said. We will write it in a clear manner in the revised text.

**Change**.Pag. 3, Line 28-30.

9. On p.4, ll.25-28, the symbols Na etc. are used to denote event indices, but ratherresemble window sizes. Using different symbols might help avoiding possible confusionwith standard notions of other papers.

**Response**.We decided to delete this sentence as it does not add extra information. The procedure of the sliding window is already described just before that.

**Change**.Pag. 5,Lines 5-7.

10. The motivation for the selection of ACC and small-world index is not clear. Insteadof the small-world index (which should be accompanied by the original reference), itwould make more sense to study ACC and APL, since both are independent whileACC and small-world index are not.

**Response**.The small-world index is an important statistical measure that reveals how a network structure deviates from random ones that account for regular seismicity. Thus, it can be used to characterize phase transitions that mark the onset of relatively big changes in the underlying topology. The original definition comes from Humphries et al. 2006 and gives an efficient way of characterizing all together the structure of the network with respect to the relation between clustering coefficient and path length.

**Change**. Pag. 5, Lines 18-21.

11. The statement that the network properties are obtained "by averaging the corresponding properties over all the nodes of the network" (p.5, l.1) is not quite obvious forthe small-world index.

**Response**.This statement is not for the small-world index but for the path lengths and clustering. We will clarify this in the revised version of the manuscript.

**Change**. Pag. 5, Lines 14-18.

12. The term "degree" should be briefly explained at its first appearance in the text.

Network scientists know this term very well, but this is not necessarily the case forseismologists.

**Response**.We agree with the reviewer. We will define it in the revised version.

**Change**. Pag. 11, Lines 5-7.

13. The term "random regular graph" (p.5, l.30) does not exist – you have either arandom or a regular graph.

**Response**.Allows us to say that the term "random regular graph" exists and refers to random graphs with uniform degree (see e.g. https://en.wikipedia.org/wiki/Random_regular_graph). However we agree that it is not a common terminology to the non-expert in the field of complex networks readers. Hencewerefer only to random networks

**Change**.Pag. 6, Line 15.

14. Can one motivate the idea that "hubs that could serve as potential epicenter locators"(p.6, ll.14-15) from known seismological principles?

**Response**.p.6 l. 24 ….node related to the mainshock epicenter. This observation along with the drop of the b-value (e.g. Papadopoulos et al., 2010) indicates stress increase close to the mainshock epicenter, thus underlying the physical link between the centralization of the BC distribution and the mainshock nucleation process. The BC of all other nodes do not change their values significantly as we present in the manuscript.

**Change**. Pag.7, Lines 15-18.

15. The authors state that "the recognition of the seismicity anomaly by topologicalmeasures does not discriminate the seismicity style, i.e., foreshocks, swarms or aftershocks"(p.8, ll.18-19). This is surely correct for the present network

constructionapproach. In turn, other recent types of construction mechanisms have been used fordeclustering earthquake catalogs and, thus, identifying fore- and aftershock sequences

(Jimenez et al., EPJST, 2009). It appears reasonable to add a corresponding comment.

**Response**.The recognition of the seismicity anomaly by topological measures, however, does not discriminate the seismicity style, i.e. foreshocks, swarms or aftershocks. The foreshock nature of the anomaly was detected only by the a posteriori knowledge that the earthquake sequence concluded with a mainshock. In seismology several branching type models have been used to identify space-time clusters, e.g. the epidemic-type aftershock sequence (ETAS) model (OGATA, 1998). Although it was tested for analyzing clustering features of foreshocks (e.g. ZHUANG and OGATA, 2006) no standard method has been introduced so far for the foreshock recognition beforehand. This is also valid for the several techniques introduced for declustering earthquake catalogues (e.g. Jimenez et al., EPJST, 2009). However, the classic seismicity statistics yields possibilities for such discrimination beforehand thanks to the discriminatory power of the b parameter which drops significantly during foreshocks. The drop of b is not only of statistical but also of geophysical value. Therefore, investigating for an equivalent discriminatory topological measure is a challenge.

**Change**. Pag. 10, Lines6-10.

16. For the definition of the small-world effect in complex networks, both ACC and APLare commonly taken into account together. What the authors report on p.9, l.17, forthe behavior of APL seems not to fully comply with the common view. I recommendconsulting the seminal work by Watts and Strogatz (Nature, 1998) for details.

**Response**.The small-world index was first introduced by Humphries et al. 2006 http://www.ncbi.nlm.nih.gov/pmc/articles/PMC1560205/ and we use the same definition as appears in that paper. In line 17 we say that the APL for small world network is of the same order as the equivalent random structure. This is in line with the definition and properties of the APL (see Watts, 1999). http://www.cc.gatech.edu/~mihail/D.8802readings/watts-swp.pdf). Actually pure random graphs exhibit relatively small average-path lengths.

However, indeed this is not so clear we will clarify this issue and cite the appropriate papers.

**Change**.Appendix

17. It is not clear what the authors mean by "local [network] property" (p.9, l.24). BCis clearly a node property, however, its computation requires global linkage informationon the entire network. In this regard, the term "local property" might be misleading.

**Response**.We agree that could be misdealing but this is the standard terminology used in the field. We will clarify better in the revised version.

**Change**. Appendix

18. P.9, ll.24-25: "BC(i) indicates that the i[-th] node acts as a central node influencingmost of the other nodes" – how has this influence to be understood in the context ofthe considered earthquake networks?

**Response**.It means that this node (patch of land) acts as a hub. This means that the seismicity revolves around this particular point. We will explain better in the revised version.

**Change**. Appendix

19. In Fig. 2c, an additional horizontal line at b=1 might help visualizing the differences for different time windows as discussed by the authors. In the caption, the right panels of Fig. 2b should be mentioned (even though they only represent a zoom of parts of he right panels).

**Response**.We will make these changes as suggested.

**Change**. A new yellow horizontal line at b=1 is added in figure 2c and a comment for the right panel is added to the figure caption

20. The cumulated BC (CBC*) is not clearly defined in the text, and its definition andmeaning are not clear from the text. This aspect needs to be clarified.

**Response**.We will describe it better in a clear manner in the revised version.

**Change**. Pag.7, Lines 12-13.

21. I recommend including Fig. S1 in the main text. A supplementary information forjust a single figure does not seem necessary.

**Response**.We agree and we will include it in the main text.

**Change**. Figure 8 instead of figure S1 is inserted in the main text

22. Throughout the manuscript, there seem to be numerous (yet minor) languageissues like confusion of singular and plural forms, use of articles, word order, etc.

Careful proofreading is recommended prior to resubmission.

**Response**.We will correct all these issues.

**2nd reviewer**

**1. Major point**

As I understand, the network analysis does not take into account the different magnitudes of the foreshocks. This means that it could depend on the chosen minimum magnitude of the dataset (now Mc=1.3). I think this is the weakest point of this kind of analysis. Results from some other different choices should be shown, in order to prove that the choice is not critical.

**Response**. The selection of the minimum magnitude of the data set was not done ad-hoc. The earthquake catalogue was tested for data completeness on the basis of the G-R diagram as done in the classical statistics.

This is an important task as mishandling data may subsequently lead to wrong evaluations for a series of important features, such as seismic rate changes, with implications for the identification of earthquake sequences, e.g. aftershocks.

Indeed the above is not highlighted enough in the text. We will explain it better in the revised version.

**Change**. Pag.4, Lines 1-3.

**2. Secondary points**

- I suppose the selection of the seismic events is limited by the depth so considering only shallow earthquakes, but this is not said. Could you please specify it? And, how much critical is the choice (e.g. showing results for two other depth choices)?

**Response**. We have considered the earthquake catalogue of the area without threshold in the focal depth.

However, the majority (nearly 97%) of the events have focal depth less than 30 km, which means that they are shallow.

The remaining may have depth up to about 45 km but we did not remove them from the catalogue allowing for some error to be involved in the depth determination. As a consequence, the data set that we used practically represents the shallow seismogenic layer.

We will write it explicitly in the revised version.

**Change**. Pag.4, Lines 5-9.

- I believe that some passages of the manuscript would be strengthen by adding some proper references that are now missing. I can suggest some (but the Authors can add alternatives), as the following: Pag.2 line 8. After "(e.g. Bufe and Varnes, 1993)" I would add the sentence: "A recent revision of the method has been proposed in order to cope with some previous limitations (De Santis et al., 2015)."

Pag.3 line 14. I would add at the end of the present sentence the following: "However, around a year before the mainshock possible effects due to fluid migration was found from magnetic data analyses (Cianchini et al., 2012)."

**Response**.We agree and we will add the appropriate references in the revised manuscript.

**Change**. Pag.2, Line 9-10& Pag.3 Line 14-15.

- Pag. 7 line 23. After "its occurrence" I would insert the following: "(look also De Santis et al. 2015)".

**Response**.We agree. We will make the necessary change as suggested.

**Change**. Pag. 9, Line 7.

- Pag.8 line 1. Please, after "2010" insert: "; De Santis et al., 2011".

**Response**.We agree and we will do so.

**Change**. Pag.9, Line 19.

- Fig.2 caption, pag. 16 line 13. When you write "2σconfidence intervals" do you really mean

"±2σconfidence intervals" or "±σ confidence intervals"? Please clarify.

**Response**.We mean "±2σ confidence intervals. We will clarify this in the revised version.

**Change**. Caption of figure 2

- Fig. 5 pag. 19 and Fig. 6 pag. 20. Although you already provide the spatial distribution of the earthquakes in Fig. 1, I believe that these Figures would be clearer if associated with the progressive spatial distribution of earthquakes in each frame, even provided in a separate Figure (if the points decrease clarity in reading).

**Response**.We have now added a new figure (Figure 9) illustrating snapshots of the seismic networks overlaid on the BC measure.

**Change**. Please see new Figure 9.

**Minor points**

**Response**.We agree and we will fix all of them in the revised version.

**Change**. Done

**References**

**Response**.We will add the suggested references

**Change**. Done

[revised manuscript text omitted]

---

## Author Response (AR2)

**Nonlinear Processes in Geophysics**

June 8, 2016

Dear Editor,

Thank you for your e-mail of 05.07.2016 informing us about the tentative acceptance of our manuscript entitled

" Foreshocks and Short-Term Hazard Assessment to Large Earthquakes using Complex Networks: the Case of the 2009 L'Aquila Earthquake".

We have now revised our manuscript including the suggested references which are indeed close enough to our work. We would like to thank you for pointing them out.

We hope that you will now consider the manuscript acceptable for publication in **Nonlinear Processes in Geophysics**.

We are looking forward to hearing from you,

Sincerely,

Constantinos I. Siettos, Ph.D.
Associate Professor

[revised manuscript text omitted]